# Structural and regulatory insights into the glideosome-associated connector from *Toxoplasma gondii*

Amit Kumar[1], Oscar Vadas[2], Nicolas Dos Santos Pacheco[2], Xu Zhang[1], Kin Chao[1], Nicolas Darvill[1], Helena Ø Rasmussen[3], Yingqi Xu[1], Gloria Meng-Hsuan Lin[2], Fisentzos A Stylianou[1], Jan Skov Pedersen[3], Sarah L Rouse[1], Marc L Morgan[1], Dominique Soldati-Favre[2]*, Stephen Matthews[1]*

[1]Department of Life Sciences, Imperial College London, London, United Kingdom; [2]Department of Microbiology and Molecular Medicine, Faculty of Medicine, University of Geneva, Geneva, Switzerland; [3]Interdisciplinary Nanoscience Center (iNANO) and Department of Chemistry, Aarhus University, Aarhus, Denmark

**Abstract** The phylum of Apicomplexa groups intracellular parasites that employ substrate-dependent gliding motility to invade host cells, egress from the infected cells, and cross biological barriers. The glideosome-associated connector (GAC) is a conserved protein essential to this process. GAC facilitates the association of actin filaments with surface transmembrane adhesins and the efficient transmission of the force generated by myosin translocation of actin to the cell surface substrate. Here, we present the crystal structure of *Toxoplasma gondii* GAC and reveal a unique, supercoiled armadillo repeat region that adopts a closed ring conformation. Characterisation of the solution properties together with membrane and F-actin binding interfaces suggests that GAC adopts several conformations from closed to open and extended. A multi-conformational model for assembly and regulation of GAC within the glideosome is proposed.

**\*For correspondence:**
Dominique.Soldati-Favre@unige.ch (DS-F);
s.j.matthews@imperial.ac.uk (SM)

## Editor's evaluation

The authors describe the first full–length crystal structure and solution conformation of the glideosome–associated connector (GAC) protein from *Toxoplasma gondii*. The data are convincing and support a model in which GAC uses multiple conformations and lipid–binding surfaces. This study is an important step towards a mechanistic understanding of glideosome assembly and function during the invasion process.

## Introduction

Cellular migration is an essential process that plays important roles in morphogenetic movements, immune cell trafficking, wound healing, and invasion. Interactions between cells and their environment are essential for the transmission of intracellular forces to the extracellular matrix. In multicellular eukaryotes, cell-cell adhesion ensures tissue integrity while providing footholds for the migration of cell within tissues (*De Pascalis and Etienne-Manneville, 2017*). Cadherins and integrins are major examples of such adhesive molecules that are coupled to the actin cytoskeleton via intracellular bridging components, such as the catenins, vinculin, and talin (*Bachir et al., 2017*).

Pathogenic organisms also exploit adherent junctions to facilitate movement with respect to host cells and invasion. *Toxoplasma gondii* is an obligate intracellular unicellular parasite and a prominent member of the Apicomplexa phylum (*Kim and Weiss, 2004*), which also includes Plasmodium, the

causative agent of human malaria (*Su et al., 1995*). These parasites share a common set of specialised apical organelles, the micronemes and rhoptries, that are critical for invasion (*Carruthers and Sibley, 1999*). The sequential secretion of both organelles leads to the formation of a moving junction (MJ) formed between the parasite and host cell plasma membranes that participates in active penetration. Apicomplexan parasites also actively egress from infected host cells and migrate across biological barriers. Substrate-dependent, forward parasite propulsion is known as gliding motility and powered by a multiprotein structure referred to as the glideosome. A myosin motor comprising myosin A (MyoA), a class XIV myosin heavy chain and its myosin light chains, together with glideosome-associated proteins (GAPs) interact with and generate rearward translocation of actin filaments (F-actin) along the parasite (*Powell et al., 2018*).

The glideosome-associated connector (GAC) protein is a central bridging component of the gliding machinery (*Jacot et al., 2016*). This large protein composed of numerous armadillo repeats (ARMs) is highly conserved throughout the Apicomplexa phylum and links F-actin to the TRAP/MIC family of surface adhesins at the plasma membrane, which targets host cell ligands and mediates adherent anchor points. A key member of the TRAP/MIC family is the transmembrane micronemal protein 2 (TgMIC2) that supports gliding motility and host cell invasion (*Huynh and Carruthers, 2006*). The ectodomain of TgMIC2 comprises a von Willebrand factor A domain and multiple thrombospondin repeat domains that associate with an accessory protein (M2AP) and host cells (*Huynh et al., 2015*; *Jewett and Sibley, 2004*; *Song and Springer, 2014*; *Tonkin et al., 2010*). TgMIC2 is predicted to connect the parasite actomyosin system through an interaction between GAC and its cytoplasmic tail. GAC moves dynamically with the MJ from the parasite apical to the basal pole during gliding motility, host cell egress, and invasion. Rearward translocation of adhesins anchored to both the parasite plasma membrane and the host membrane by the inner-membrane-associated glideosome generates parasite forward movement (*Carruthers and Tomley, 2008*; *Frénal et al., 2017a*).

The initial apical location of GAC depends on the activity of an apical lysine methyltransferase (AKMT), through a yet unknown mechanism (*Jacot et al., 2016*). Recently ultrastructure expansion microscopy localised GAC and formin 1 (FRM1) to the preconoidal rings (PCRs; *Dos Santos Pacheco et al., 2022*). Importantly, FRM1 is the only and essential nucleator of actin polymerisation to drive conoid extrusion and parasite motility and invasion (*Tosetti et al., 2019*). As part of the conoid, the PCRs serve as platform for the assembly of the glideosome. Membrane association of GAC relies on its capacity to bind phosphatidic acid (PA), an essential lipid mediator for microneme secretion which assists in the correct engagement of GAC (*Bullen et al., 2016*). Despite significant efforts, high-resolution structural insight into GAC and its multiple interactions has not been available since its discovery. An initial small-angle X-ray scattering (SAXS) model predicted an elongated club-shaped conformation with the C-terminal Pleckstrin homology (PH) domain lying at the extremity of the structure (*Jacot et al., 2016*).

Here, we describe combined X-ray crystallography, nuclear magnetic resonance (NMR), SAXS, and hydrogen/deuterium exchange coupled to mass spectrometry (HDX-MS) with course-grained molecular dynamics (CG-MDs) analyses to illuminate the structure and conformations adopted by full-length GAC from *T. gondii*. Structure validation by biochemical and parasite assays provides insights into membrane and actin binding, ultimately allowing us to propose a model for assembly within the glideosome.

## Results

### The structure of full-length TgGAC

Well-diffracting and reproducible crystals were obtained at pH 5 for native full-length TgGAC (residues 1–2639; *Kumar et al., 2022*). Selenomethionine-substituted crystals were also obtained that produced sufficient anomalous signal for phase determination. The structure was solved by multiple-wavelength anomalous dispersion (MAD) to 2.7 Å resolution with an $R_{free}$ of 26% (*Table 1*). The electron density for residues 7–2504 was of sufficient quality to facilitate modelling for these residues (*Figure 1* and *Figure 1—figure supplement 1A*). The remaining electron density showed evidence of peptide backbone but could not be confidently modelled and refined, indicating a degree of conformational flexibility for residues 2505–2639.

**Table 1.** Data collection, phasing, and refinement statistics.

| | TgGAC | | |
|---|---|---|---|
| Space group | P 21 21 21 | | |
| Cell dimensions | | | |
| $a$, $b$, and $c$ (Å) | 119.078, 123.605, and 221.508 | | |
| α, β, and γ (°) | 90, 90, and 90 | | |
| | Peak | Edge | Remote |
| Wavelength (Å) | 0.9795 | 0.9796 | 0.9722 |
| Resolution range (Å) | 110.75–2.67 (2.72–2.67) | 110.75–2.67 (2.72–2.67) | 110.75–2.67 (2.72–2.67) |
| Total number of reflections | 904,519 | 750,918 | 798,669 |
| Unique reflections | 92,922 | 76,602 | 81,143 |
| $R_{pim}$ | 0.036 (0.48) | 0.032 (0.517) | 0.031 (0.504) |
| $I/\sigma I$ | 14.00 (0.8) | 15.8 (0.73) | 14.70 (0.71) |
| Completeness (%) | 100 (100) | 100 (100) | 100 (100) |
| Multiplicity | 9.7 (9.7) | 9.7 (10.7) | 9.8 (10.3) |
| R-merge | 0.108 (1.627) | 0.096 (1.64) | 0.093 (1.44) |
| R-meas | 0.114 (1.698) | 0.101 (1.72) | 0.098 (1.528) |
| CC1/2 | 0.99 (0.58) | 0.999 (0.57) | 0.99 (0.65) |
| Refinement statistics | | | |
| No. of reflections for refinement | 92,300 (9069) | | |
| No. of reflections for $R_{free}$ | 4502 (409) | | |
| $R_{work}$ | 0.2095 (0.4036) | | |
| $R_{free}$ | 0.2683 (0.3990) | | |
| CC (work) | 0.904 (0.508) | | |
| CC(free) | 0.910 (0.440) | | |
| No. of non-hydrogen atoms | 18,793 | | |
| Macromolecules | 18,679 | | |
| Solvent | 114 | | |
| Ramachandran favoured/allowed/outliers (%) | 93.4/6.1/0.45 | | |
| Average B-factor | 76.88 | | |
| Macromolecules | 70.92 | | |
| Solvent | 70.07 | | |
| Rms bond lengths (Å) | 0.009 | | |
| Rms bond angles (°) | 1.47 | | |
| Clashscore | 17.98 | | |

Values in parentheses are for highest-resolution shell.
Data from a single crystal were used to solve the structure.

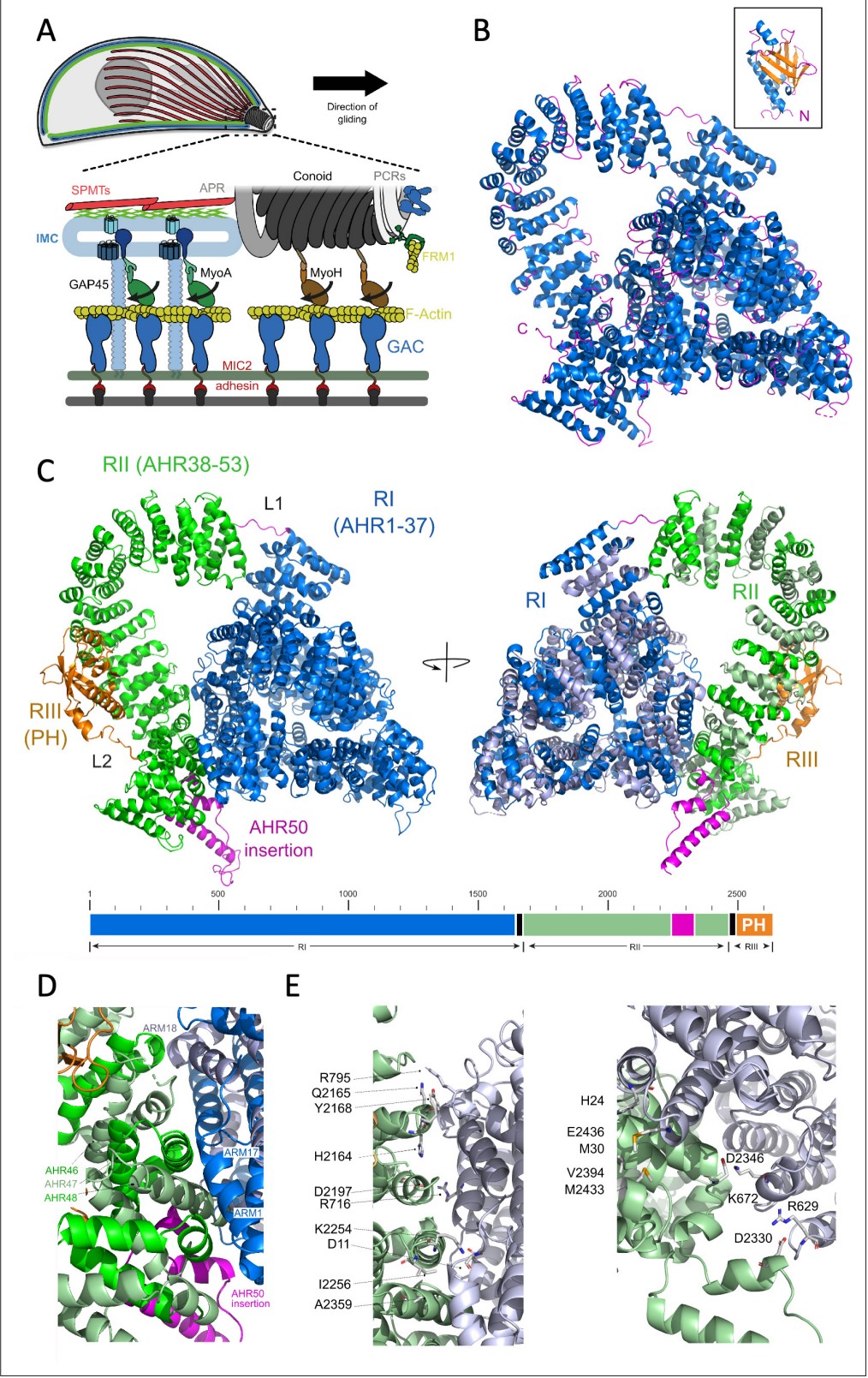

**Figure 1.** The organisation and structure of TgGAC. (**A**) Schematic diagram of the apicomplexan glideosome. Key molecular components and structures are indicated, including subpellicular microtubules (SPMTs), inner membrane complex (IMC), preconoidal rings (PCRs), apical polar ring (APR), Formin-1 (FRM1), filamentous actin (F-actin), glideosome-associated protein 45 (GAP45), myosin motors (MyoA and MyoA), microneme protein 2 (MIC2), and

*Figure 1 continued on next page*

*Figure 1 continued*

glideosome-associated connector (GAC) protein. (**B**) Crystal structure of TgGAC$_{7-2504}$ and representative nuclear magnetic resonance (NMR) model of TgGAC$_{2505-2639}$ (inset). Helices shown in blue, β-strands in orange and loops in magenta. (**C**) Full-length TgGAC model comprising the combined crystal structure and NMR-validated AlphaFold2 structures in (**B**). Region I (R1: residues 1–1665) comprising the first 37 consecutive armadillo (ARM)/HEAT-like repeats (AHRs) is supercoiled into a three-layer pyramid structure (blue). AHR region II comprising 16 ARM repeats (RII: AHR38-53 residues 1670–2489) forms the superhelical arch (green). The C-terminal PH domain encompassing 2511–2639 of RIII is shown in orange (left). Second orientation related by a 180° rotation with even numbered AHRs shown in grey and odd number shown in blue for RI and green for RII. The R1–RII linker and AHR50 which has a helix-loop-helix insertion are shown in magenta. (**D**) The N/C interface between RI and RII showing key interacting AHRs. (**E**) Key residues specific interactions across the N/C interface. Cartoon representations coloured in light blue for RI and light green for RII.

The online version of this article includes the following source data and figure supplement(s) for figure 1:

**Source data 1.** Maps and coordinates.

**Figure supplement 1.** Assessing the quality of structural data for glideosome-associated connector (GAC).

**Figure supplement 1—source data 1.** Nuclear Overhauser effect (NOE) assignment, nuclear magnetic resonance (NMR) structure calculation data, PDB file for TgGAC PH domain.

**Figure supplement 2.** Secondary structure for the crystal structure of TgGAC.

**Figure supplement 3.** Structural homology with TgGAC.

**Figure supplement 4.** Sequence alignment for the crystal structure of TgGAC.

To gain insight into the structure of this C-terminal region, we initiated a solution NMR spectroscopy approach with a construct encompassing residues 2505–2639 (TgGAC$_{2505-2639}$). NMR spectra confirmed an independently folded domain, and over 80% of the backbone resonances could be confidently assigned (***Figure 1—figure supplement 1B***). Several amide peaks were broadened or absent from the spectra due to conformational exchange in some loop regions. Using available nuclear Overhauser effect spectroscopy (NOESY) and assignment spectra an automated NMR structure calculation was performed using CYANA (***Güntert and Buchner, 2015***) implemented within the ARTINA approach (***Klukowski et al., 2022***). The final NMR ensemble was calculated based on 3533 nuclear Overhauser effect and superposes with an average pairwise backbone root mean square deviation (RMSD) of 0.17 Å (***Figure 1—figure supplement 1C***). Structure statistics are shown in ***Table 2***. We also independently generated a structural model for TgGAC$_{2505-2639}$ using AlphaFold2 (***Jumper et al., 2021***; ***Varadi et al., 2022***). Excellent agreement is observed between the experimental NMR structure and predicted from the AlphaFold2 structure with a backbone RMSD of 1.5 Å (***Figure 1—figure supplement 1D***). The precision of the NMR ensemble is lower between residues 2548 and 2560 due to missing resonance assignments within this region.

The structure of TgGAC$_{2505-2639}$ adopts a PH-like domain fold comprising a seven-stranded β-barrel with three α-helices. The presence of an extended N-terminal-helix is reminiscent of the PH domains from TgAPH and TgISP (***Darvill et al., 2018***; ***Tonkin et al., 2014***) and appears to be a common feature among apicomplexan PH domains. To provide an illustrative model for full-length TgGAC, the validated structure of TgGAC$_{2505-2639}$ (TgGAC$_{PH}$) was positioned in the available electron density map for N-terminus of this region and a linker modelled with MODELLER (***Fiser et al., 2000***).

The crystal structure of TgGAC$_{7-2504}$ comprises 169 helices (***Figure 1*** and ***Figure 1—figure***

**Table 2.** Nuclear magnetic resonance (NMR) structure calculation statistics.

| 20 lowest energy structures | TgGAC$_{PH}$ |
| --- | --- |
| Assigned nuclear Overhauser effect (NOE) peaks | |
| Intra-residue | 1648 |
| Medium range ($|i - j| \leq 4$ | 640 |
| Long range ($|i - j| > 4$ | 1260 |
| Total | 3548 |
| Average CYANA target function value | 12.4 |
| Ramachandran plot | |
| % in most favoured positions | 79.7±0.7 |
| % in allowed regions | 20.2±0.4 |
| % in generously allowed | 0.2±0.1 |
| % in disallowed regions | 0.0±0.0 |
| Atomic coordinates | |
| Pairwise backbone RMSD secondary structures (Å) | 0.17±0.01 |

*supplement 2*) arranged in 53 consecutive helical bundles that resemble ARM and HEAT-like repeats (AHRs; *Kippert and Gerloff, 2009*) linked to the mixed α/β C-terminal PH domain (*Figure 1A*). The architecture can be divided into three major regions (RI, RII, and RIII). RI (residues 1–1665) comprises 37 consecutive AHRs of approximately 40 residues long that are supercoiled into a three-layer pyramid structure (*Figure 2A*). The final ARM of region I (AHR 37) is extended with a six amino acid linker (L1) to start a second large AHR region (RII). RII comprises 16 canonical armadillo repeats (AHR38-53) and forms a superhelical arch that contacts the base of the RI pyramid helical regions. This interface encompasses a surface area of 1550 Å², with AHR1 and AHR17-18 from RI forming one side while AHR48-50 from RII form the other (*Figure 1C*). The interface comprises several prominent electrostatic interactions (K2254-D11, E2436-H24, D2330-R629, D2346-K672, and D2197-R716; *Figure 1D*). AHR50 also has a large helix-loop-helix insertion (residues 2278–2337) between the second and third helices which creates a prominent protrusion that also stabilises the interface (*Figure 1B and C*). A basic AHR53 completes the region, (II) and an ordered linker from 2489 to 2510 (L2) extends to the adjacent C-terminal PH domain which comprises region III (*Figure 1A and B*). Residues forming these key features of the structure are highly conserved across GAC orthologues in Apicomplexa (*Figure 1—figure supplement 3*).

## GAC adopts multiple extended conformations in solution

The closed conformation observed in the TgGAC crystal structure deviates significantly from the extended club-shaped structure that was proposed from previous SAXS analyses (*Jacot et al., 2016*), which raises concern that it could be an artefact of crystallisation. To shed light on this, we examined AlphaFold2 structures predicted for full-length TgGAC and PfGAC (*Jumper et al., 2021*; *Varadi et al., 2022*). Remarkably, the overall architecture of the predicted structures is very similar to that observed in the crystal structure of TgGAC (*Figure 2A*), as it is characterised by a series of tandem AHRs that are supercoiled into a large ring and is closed by specific interactions between N- and C-terminal AHRs. While the interface in the predicted TgGAC and PfGAC is partially separated (*Figure 2A*), the interacting regions overlap with those identified experimentally for TgGAC. The N-/C-terminal interface that characterises the closed structure is conserved at the amino acid sequence level (*Figure 2B* and *Figure 1—figure supplement 3*).

We next performed a SAXS analysis over a range of solution conditions to determine whether an open structure is indeed present in solution. GAC was measured at a concentration of 1.3 mg/mL as a function of pH from 4.0 to 8.0. We chose this concentration as a compromise between having the lowest concentration to avoid intermolecular interactions leading to aggregation whilst maintaining good signal to noise. The largest difference is observed as an increase in intensity at low scattering vector moduli, *q* (*Figure 2B*), where the intensity is proportional to the mass of the protein/complexes, suggesting that oligomerisation/aggregation is occurring and is most pronounced at pH 5.0. This is consistent with SEC profiles at pH 5 and 8 showing significantly earlier elution of TgGAC at pH 5 (*Figure 2—figure supplement 1A*). Furthermore, the addition of salt has no effect of the elution volume at both pHs.

To explore the structure of TgGAC in solution at pH 8.0, the mass and size were determined both by a Guinier fit (*Figure 2—figure supplement 1B*) and using an indirect Fourier transform (IFT) routine for calculating the pair distance distribution function, *p(r)* (*Figure 2C*). Note that due to the oligomerisation/aggregation at low pH, the overall size of the complex is too large to be resolved by our instrument setup, as the slope of the intensity at low *q* is too large. When the overall size cannot be established, the mass and *p(r)* functions cannot be reliably calculated. At pH 8.0, masses of 264 kDa (Guiner) and 286 kDa (IFT) were obtained, which are close to the expected value of 286 kDa for a TgGAC monomer. The radius of gyration, $R_g$, values of, respectively, 105 Å and 122 Å were obtained, which are much larger than the $R_g$ of 48 Å calculated from the crystal structure of TgGAC. This discrepancy in size is also clearly seen when comparing the *p(r)* function calculated from the SAXS data and the crystal structure, where the maximum intramolecular distances are 394 Å and 138 Å, respectively (*Figure 2C*). The theoretical scattering curve can be calculated from the coordinates of the full-length TgGAC structure and compared to the measured data (*Figure 2D*). It shows that at medium to large *q*, the structures are relatively similar, indicating that they contain similar structural components on shorter length scales, but at medium to low *q*, there is a large difference showing that the overall size and shape are very different. To explore this further, an ab initio model was build using the ATSAS

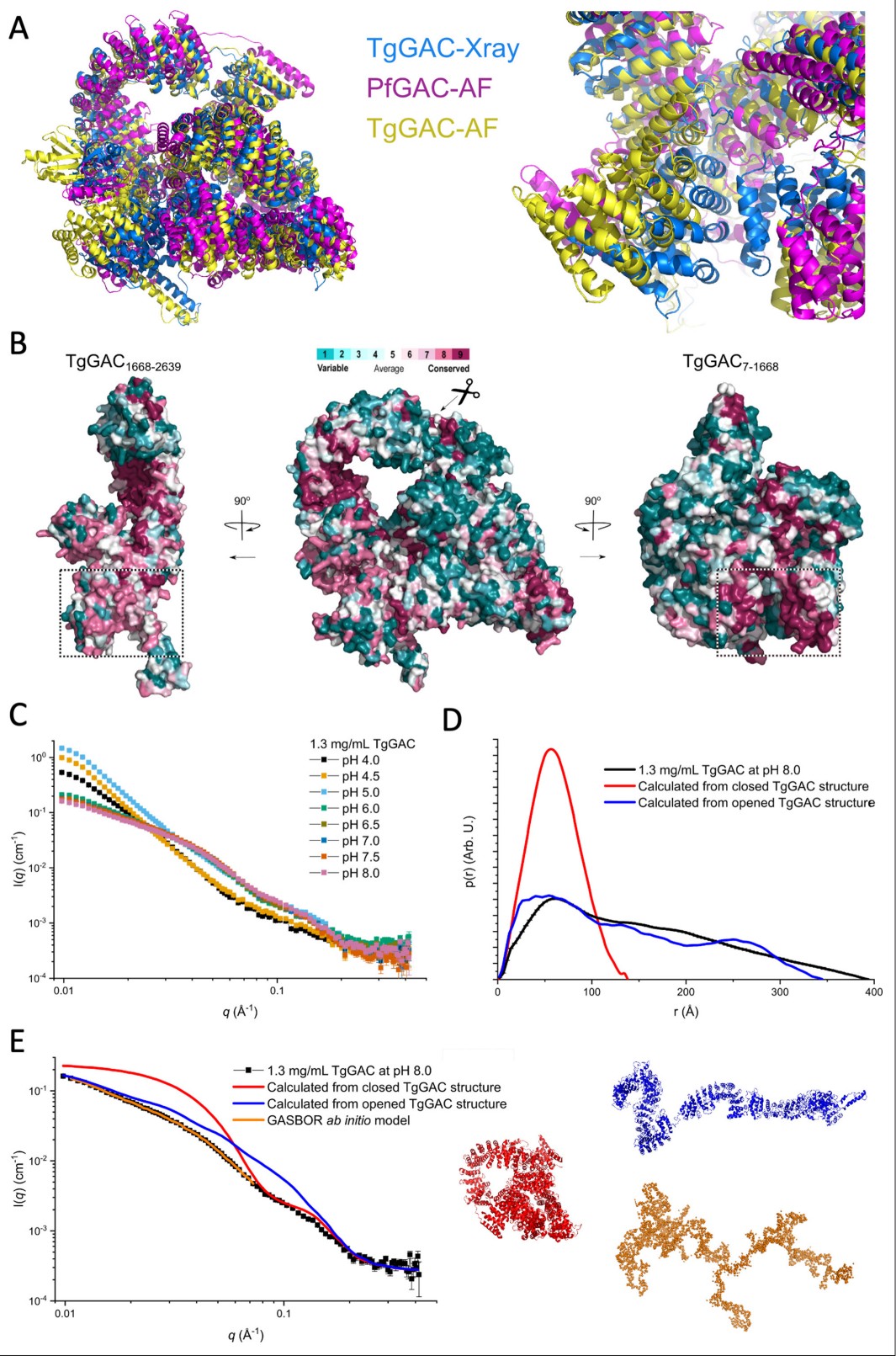

**Figure 2.** Glideosome-associated connector (GAC) adopts both open and closed structures. (**A**) Superimposition of the experimentally determined composite structure of TgGAC with those generated from AlphaFold2 for TgGAC and PfGAC, showing that the closed structure is conserved (left). N/C-terminal interface is highly similar but partially opened in the predicted structures of TgGAC and PfGAC. The PH domain is not displayed for clarity

*Figure 2 continued on next page*

*Figure 2 continued*

(right). (**B**) Conservation profile on a surface representation for TgGAC in three different orientation generated using ConSurf (*Landau et al., 2005*). Centre – full-length TgGAC in the same conformation as (**A**). TgGAC is split in two and each half rotated 90° to display the N/C interface. C-terminal regions RII and RIII (left) and N-terminal region RI (right). The surface is coloured according to its ConSurf conservation score, which varies from cyan which are highly variable to maroon for highly conserved. (**C**) Small-angle X-ray scattering (SAXS) data for GAC at pH 4.0–8.0. (**D**) Pair distance distribution functions (*p(r)*) for GAC SAXS data at pH 8.0, calculated from full-length closed TgGAC structure and calculated for the opened structure generated by molecular dynamic (MD) simulations. (**E**) Modelling of TgGAC solution structure at pH 8.0, calculated theoretical scattering curves for the full-length closed TgGAC structure (red), and calculated for the opened full-length TgGAC structure (blue), and a GASBOR *ab initio* model (orange). Structures are shown in colours corresponding to those of the scattering curves.

The online version of this article includes the following source data and figure supplement(s) for figure 2:

**Source data 1.** Raw small-angle X-ray scattering (SAXS) data for TgGAC.

**Source data 2.** Movie of steered molecular dynamic (MD) trajectory generating an extend TgGAC conformation.

**Source data 3.** Final extended full-length TgGAC after steered molecular dynamic (MD).

**Figure supplement 1.** Size-exclusion chromatography of glideosome-associated connector (GAC) and analysis of small-angle X-ray scattering (SAXS) data at pH 8.0 with comparison to crystal structure.

program GASBOR (*Svergun et al., 2001*), which is highly extended and fits the low $q$ part of the experimental data (*Figure 2D*). A dimensionless Kratky plot also illustrates that the structure in pH 8.0 solution is much more flexible than would be predicted from the crystal structure, but also it is not completely unfolded (*Figure 2—figure supplement 1*). To investigate the conformation changes occurring at pH 8.0 compared to the crystal structure in solution, a series of extended models were generated by steered-MD by applying a pulling force between the C- and N-terminal side of the interface, followed by the subsequent separation of the supercoiled pyramid. The model of these that fits best with the data is the most extended, which maintain the domain structures (*Figure 2D*). The curve fits the data well at high and low $q$; however, it is significantly above the data in the intermediate range. This means that further loss of domain structure is likely occurring. We did not attempt to pursue this further as it would be too speculative to predict which regions lose their secondary structure. Note that the p(r) function and the Kratky plot of the scattering for the extended structure are shown in *Figure 2—figure supplement 1*. Overall, the data and the models show that even though TgGAC is primarily a monomer in solution at pH 8.0, it forms a much more extended structure than seen in the crystal structure whilst maintaining key structural elements. Collectively, it can be concluded that a closed structure is likely to be an important stable functional state as the N-/C-interface is encoded within the sequence conservation. The extended, flexible structures observed in solution may play a role is a facilitating the assembly GAC with its binding partners.

## Mapping the PA-binding interface for the C-terminal PH domain of GAC

An extended patch of conserved basic residues is formed on one face of the PH domain in GAC which is reminiscent of the charge distribution on TgAPH (*Darvill et al., 2018*), suggesting a similar role in phospholipid binding. Using phospholipid strip assays, it had been previously demonstrated that both PfGAC and TgGAC bind specifically to PA (*Jacot et al., 2016*). Building upon our earlier work on the PA-binding protein TgAPH (*Darvill et al., 2018*), we employed CG-MDs simulations to characterise the binding of TgGAC to PA within a membrane environment. In all three simulation repeats, GAC bound to the membrane within 10 µs (*Figure 3—figure supplement 1*). Analysis of protein-lipid contacts highlighted two key membrane-bound regions, which localised to the PH domain and an adjacent basic protrusion from AHR53 (*Figure 3*).

NMR titrations were subsequently employed to quantify and comprehensively map the PA-binding site experimentally. To circumvent any issues with the missing NMR resonances for TgGAC$_{PH}$, we explored the equivalent region from the GAC homologue from *Plasmodium falciparum*, which has 53% sequence identity over the PH domain (PfGAC$_{2471-2605}$) with an improved NMR spectra that enabled over 95% of the backbone residues to be assigned (*Figure 4—figure supplement 1*). First, 1D NMR binding assays were performed with PA-enriched large unilamellar vesicles (LUVs of ~100 nm diameter) to measure the affinity of the interaction. Titration with LUVs composed of 1-palmitoyl-2-o leoyl-sn-glycero-3-phosphocholine (POPC) and POPA (50:50% POPC:POPA) caused a significant loss

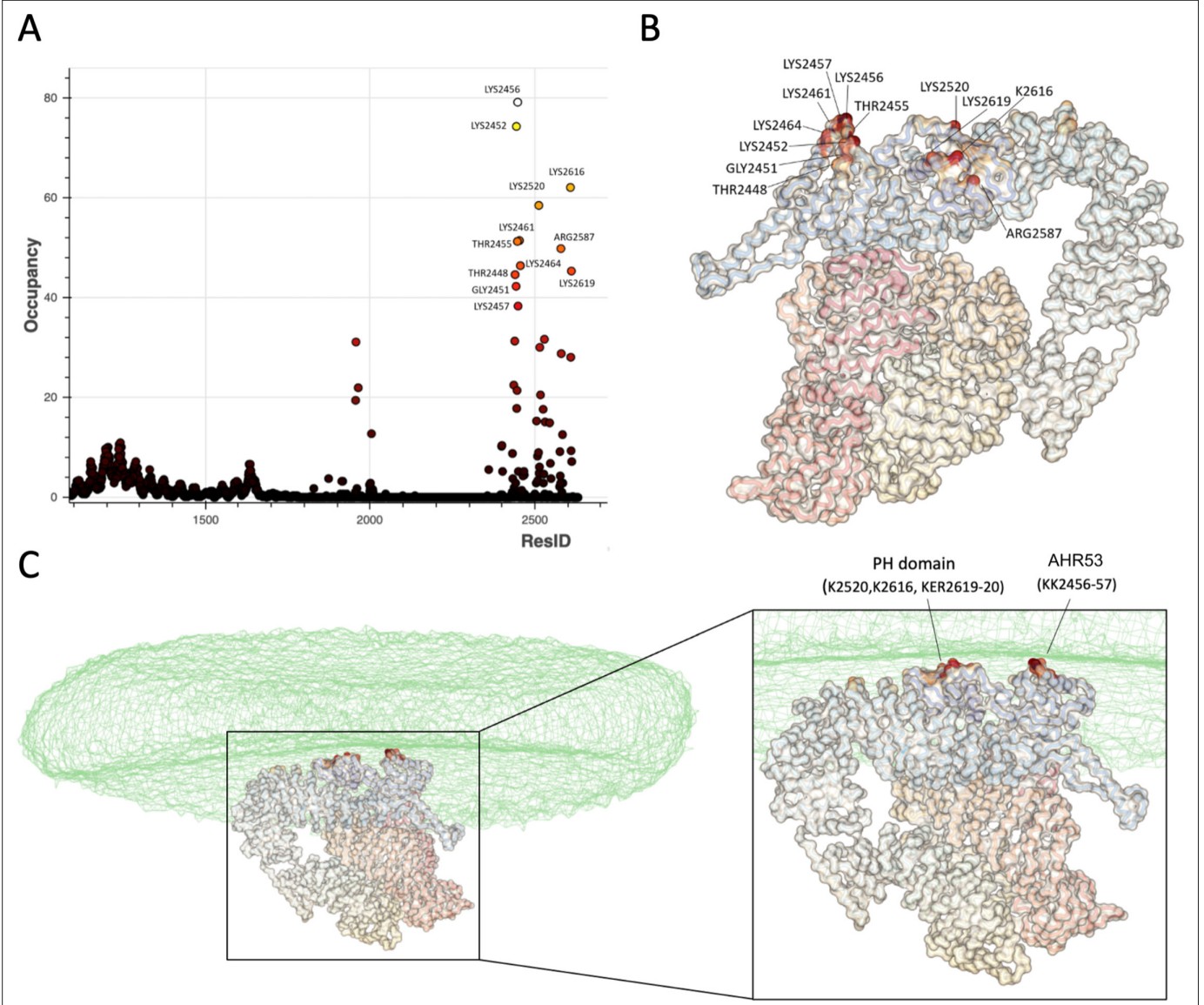

**Figure 3.** Simulation of TgGAC binding to phosphatidic acid (PA)-containing membranes. (**A**) Analysis of residues that interact with 1-palmitoyl-2-oleoyl-sn-glycero-3-phosphate (POPA) as a percentage of contact during the simulation time. Analysis was performed for the final 5 µs of the simulation. (**B**) Key residues from mapped onto the TgGAC structure. (**C**) As in (**B**) in the contact of the membrane which is displayed as a transparent surface.

The online version of this article includes the following source data and figure supplement(s) for figure 3:

**Source data 1.** Molecular dynamics trajectory analysis data for TgGAC.

**Figure supplement 1.** Molecular simulations of glideosome-associated connector (GAC) and phosphatidic acid (PA)-containing membranes.

in PfGAC$_{PH}$ NMR signals (**Figure 4A**), indicating binding to the 'NMR invisible' LUVs, whilst almost no loss in signal was observed upon titration with LUVs composed solely of POPC. Binding curves were generated from integration of the NMR signals and (**Figure 4B**) apparent dissociation constants ($Kd_{app}$) calculated (**Figure 4C**). The $Kd_{app}$ for PfGAC$_{PH}$ binding LUVs composed of 50% POPA was calculated to be 60±3 µM.

We next measured paramagnetic relaxation enhancements (PREs) using small PA-enriched MSP1D1$_{\Delta H4-5}$ nanodiscs. PA-enriched MSP1D1$_{\Delta H4-5}$ nanodiscs (40:60%, POPA:POPC) or nanodiscs containing no PA (100% POPC), doped with (paramagnetic nanodiscs) or without (diamagnetic nanodiscs) PEDTPA-Gd$^{3+}$ paramagnetic lipid, were generated. 2D $^1$H-$^{15}$N HSQC spectra were recorded for $^{15}$N-labelled PfGAC$_{PH}$ in the presence of paramagnetic or diamagnetic nanodiscs. Paramagnetic induced relaxation enhancements of membrane-interacting regions were measured from reductions

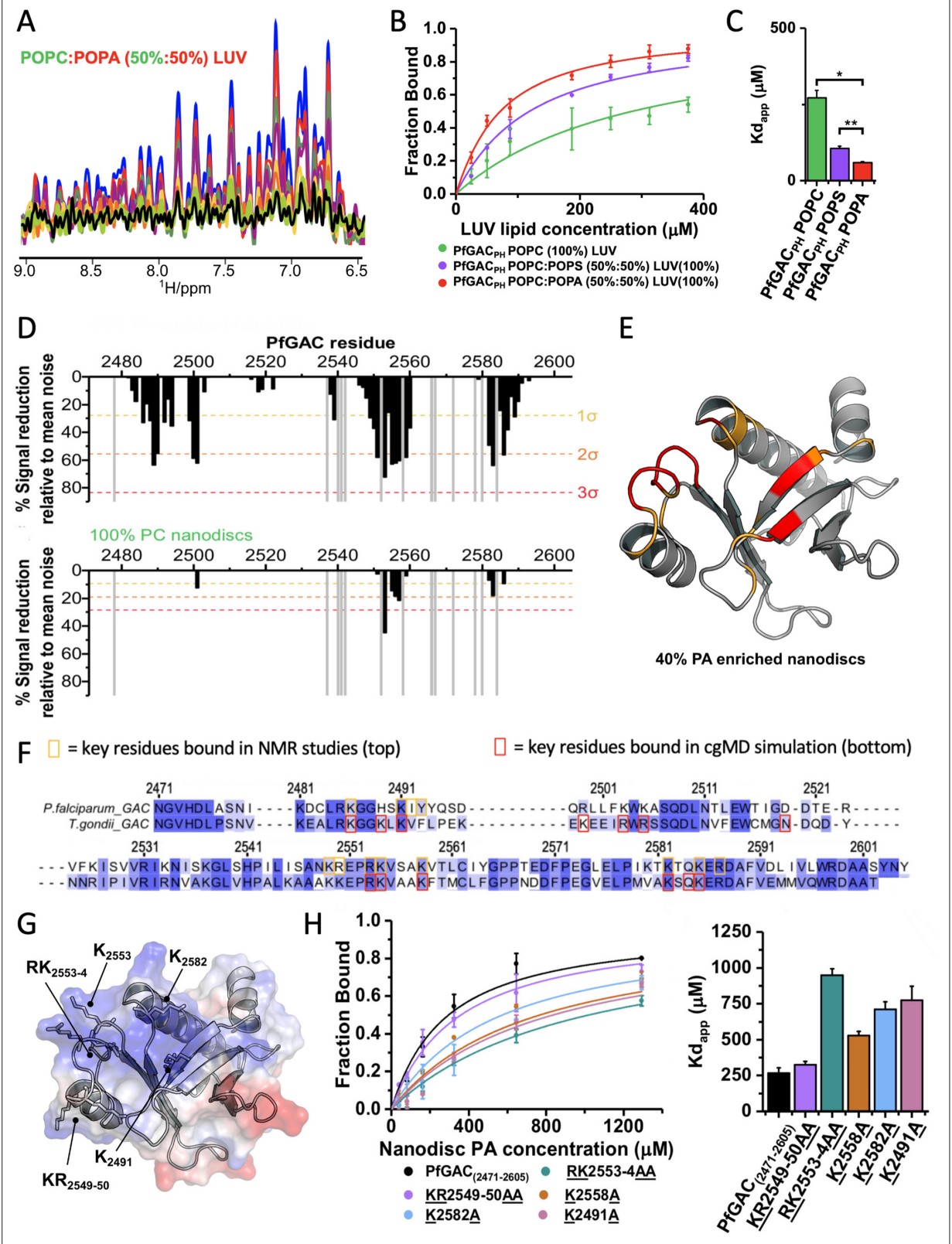

**Figure 4.** Nuclear magnetic resonance (NMR)-based binding assays for PfGAC$_{2471-2605}$ (PfGAC$_{PH}$) to phosphatidic acid (PA)-enriched unilamellar liposomes. (**A**) 1D 1H-NMR spectrum (9.4–6.5 ppm) of PfGAC$_{PH}$ upon titration with increasing concentrations of large unilamellar vesicles (LUVs) composed of POPC:POPA (50:50%) LUV molar ratios: blue, free PfGAC$_{PH}$ in solution; red 1:2; green 1:4; purple 1:7; yellow 1:15; orange 1:20; lime 1:25; black 1:30. (**B**) Binding curves generated from spectral integration and expressed as the fraction of bound protein for variable LUV compositions

*Figure 4 continued on next page*

*Figure 4 continued*

(POPC [100%] green, POPC:POPS [50:50%] purple, or POPC:POPA [50:50%] red). (**C**) Apparent dissociation constants ($K_{dapp}$) for binding LUVs were calculated from fitting binding curves. (**D**) Plot of PfGAC$_{PH}$ paramagnetic relaxation enhancements (PREs) with PA-enriched MSP1D1H4-5 nanodiscs top (POPC:POPA:PE-DTPA-Gd3+46:40:14%) and bottom (POPC:PE-DTPA-Gd3+86:14%), against sequence number. Dashed lines represent 1 (yellow), 2 (orange), or 3 (red) SD from the mean noise (baseline). (**E**) PA PREs mapped onto the structure of PfGAC$_{PH}$, residues and coloured if greater than 2 (orange) or 3 (red). (**F**) Comparison of contact resides from molecular dynamic (MD) and NMR mapped on to the sequence alignment between PfGAC$_{PH}$ and TgGAC$_{PH}$. (**G**) Electrostatic surface representation of PfGAC$_{PH}$ revealing an extensive surface patch of positive charge surface charge with key mutated residues labelled. (**H**) Binding curves of PfGAC$_{PH}$ mutants generated from 1D NMR titration with calculated $Kd_{app}$. Data are shown as mean of three replicates ±1σ. Apparent dissociation constants ($Kd_{app}$) were calculated from fitted binding curves. Data represent mean ±1σ for fitted curves.

The online version of this article includes the following source data and figure supplement(s) for figure 4:

**Source data 1.** Raw and processed nuclear magnetic resonance (NMR) data for 1D small unilamellar vescicle (SUV) titration with PfGAC$_{PH}$.

**Source data 2.** Raw and processed nuclear magnetic resonance (NMR) data for paramagnetic relaxation enhancement (PRE) measurements of PfGAC$_{PH}$ with nanodiscs.

**Figure supplement 1.** Nuclear magnetic resonance (NMR) assignments of PfGAC$_{PH}$ and lipid binding paramagnetic relaxation enhancement (PRE) data for TgGAC$_{PH.}$

**Figure supplement 1—source data 1.** Raw and processed nuclear magnetic resonance (NMR) data for SUV titration of TgGAC$_{PH.}$

**Figure supplement 1—source data 2.** Raw and processed nuclear magnetic resonance (NMR) data for paramagnetic relaxation enhancement (PRE) measurements of TgGAC$_{PH}$ with nanodiscs.

in signal intensities. Numerous PREs were observed for PfGAC$_{PH}$ upon the addition of PA-enriched paramagnetic nanodiscs (*Figure 4D* - top) but not with 100% POPC nanodiscs (*Figure 4D* - bottom).

PREs mapped onto the structure of PfGAC$_{PH}$ reveal a contiguous surface formed from residues located within the β1-strand, bordering the β1–β2 loop, within the β5–β6 loop and the loop region between β7 strand and C-terminal α-helix (*Figure 4E*). A comparative analysis of the PH binding residues compared to simulation with full-length TgGAC indicated that the PH domain alone binds with the same interface in the context of the whole GAC protein (*Figure 4F*). NMR titrations performed with TgGAC$_{PH}$ and PA-enriched paramagnetic nanodiscs confirmed an identical PA-binding surface to PfGAC (*Figure 4* and *Figure 4—figure supplement 1B*). Collectively, these results suggest that the solvent exposed positively charge surface in the GAC PH domain represents a PA-specific membrane interacting interface (*Figure 4F*). Conserved basic residue sidechains are likely to coordinate PA lipid phosphate head groups (*Figure 4G* and *Figure 1—figure supplement 3*). A series of alanine substitution mutants were then generated in PfGAC$_{PH}$ for residues identified from NMR mapping. These mutants were tested using the 1D NMR binding assay to quantitatively assess the consequence of mutations on PA-binding residues (*Figure 4H*).

Compared to WT ($Kd_{app}$ of 270±40 µM), mutation of basic residues reduced the affinity for PA-enriched MSP1E1 nanodiscs, indicating that these residues are important for binding PA within a membrane environment (*Figure 4H*). The reduction in affinity correlates with position of basic residues relative to the centre of the solvent exposed positive charges, i.e., the largest effect is observed for the RK$_{2553–54}$AA mutant ($Kd_{app}$ of 950±50 µM) compared to the KR$_{2549–50}$AA mutant ($Kd_{app}$ of 330±20 µM). The same trend is observed for K$_{2491}$A ($Kd_{app}$ of 770±100 µM) and K$_{2582}$A ($Kd_{app}$ of 710±50 µM), which are more centrally located than K$_{2558}$A ($Kd_{app}$ of 550±50 µM).

To establish whether the PA-binding interface identified through NMR analyses on PfGAC$_{PH}$ mutants display a similar deficiency in full-length TgGAC, we exploited a liposome binding assay in which bound protein was quantified by sedimentation and sodium dodecyl-sulfate polyacrylamide gel electrophoresis (SDS-PAGE). We first tested this assay for full-length TgGAC, TgGAC$_{PH}$, and the known PA sensor TgAPH (*Darvill et al., 2018*) in the presence of liposomes with no PA and liposomes containing 50% PA. All three proteins were bound specifically to PA-enriched liposomes with most of the protein present in the pellet after ultracentrifugation, whereas in the absence of PA, the proteins were found in the supernatant (*Figure 5A*). Three mutants were chosen based on the NMR data on PfGAC and TgGAC and were generated in both full-length TgGAC and isolated PH domain constructs. The first mutant focuses on the major positive-charged patch that exhibited the largest effect in NMR assays (namely RK$_{2587–88}$AA in TgGAC), the second is a triple mutation in the downstream basic region (KER$_{2619–2621}$AAA in TgGAC) and the third combines both (RK$_{2587–88}$AA/KER$_{2619–2621}$AAA in TgGAC). All mutant proteins tested exhibited a significant reduction in binding to PA enriched liposomes (*Figure 5B and C*), with the most dramatic effects observed for the triple mutant KER$_{2619–2621}$AAA

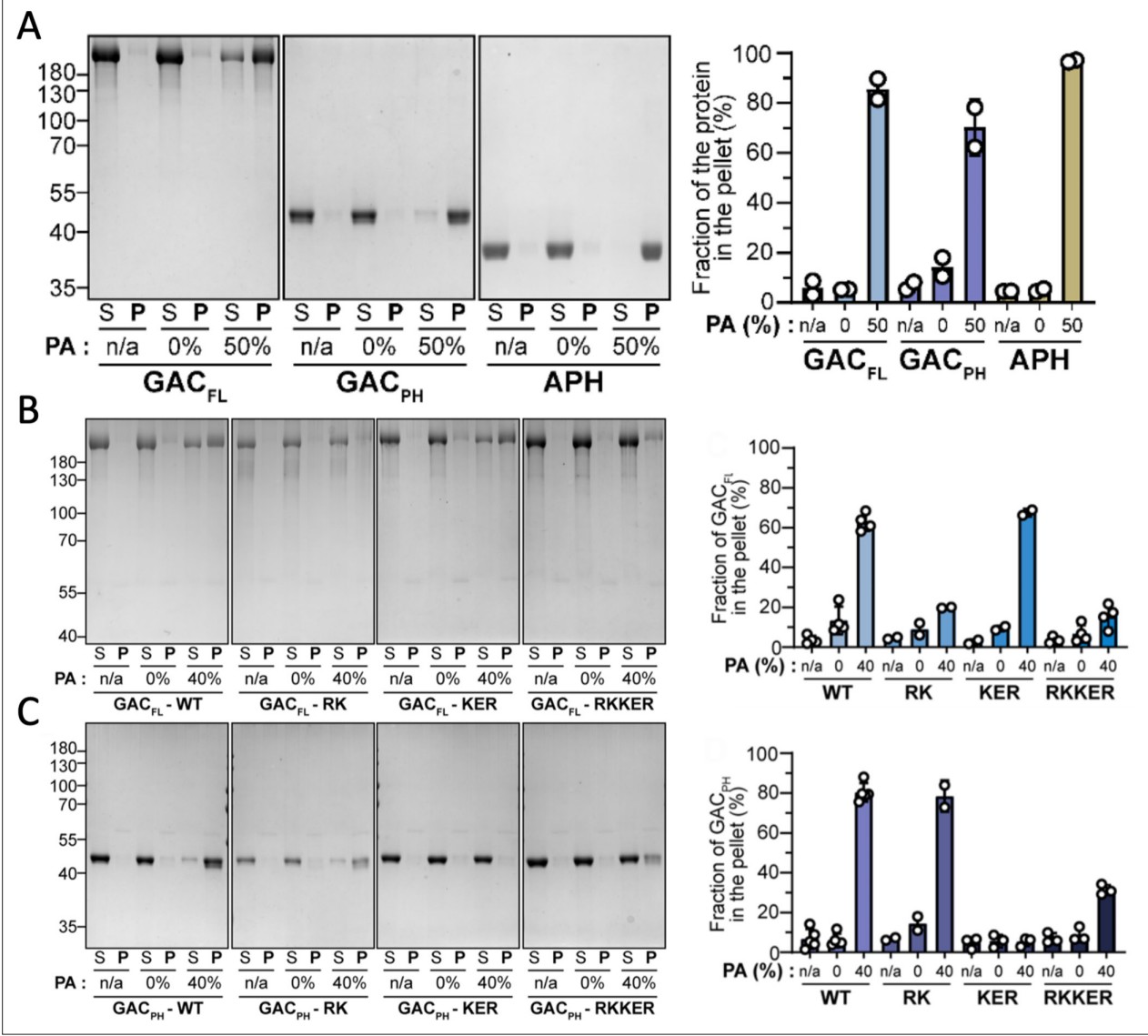

**Figure 5.** Liposome binding assays of phosphatidic acid (PA) binding by glideosome-associated connector (GAC) and PH domain mutants in vitro. (**A**) Liposome binding assay with the three proteins of interest. A representative gel stained by Coomassie blue (left). Quantification of the pellet fraction measured by band densitometry (n=2). n/a=no liposome. S=supernatant fraction after ultracentrifugation. P=pellet fraction after centrifugation (right). (**B**) Representative gel stained by Coomassie blue from liposome binding assays with full-length TgGAC$_{FL}$ and the three mutated versions (left). Quantification of the GAC$_{FL}$ pellet fraction measured by band densitometry (n=2–4; right). n/a=no liposome. S=supernatant fraction after ultracentrifugation. P=pellet fraction after centrifugation. RK = RK/AA mutations. KER = KER/AAA mutations. RKKER = RK/AA +KER/AAA mutations. (**C**) Representative gel stained by Coomassie blue from liposome binding assays with TgGAC$_{PH}$ and the three mutated versions (left). Quantification of the GAC$_{PH}$ pellet fraction measured by band densitometry (n=2–4; right).

The online version of this article includes the following source data for figure 5:

**Source data 1.** Full, uncropped SDS-PAGE images for liposome binding experiments.

**Source data 2.** Figure and caption for liposome binding assays using uncropped SDS-PAGE images with key bands labelled.

**Source data 3.** Liposome binding data tables.

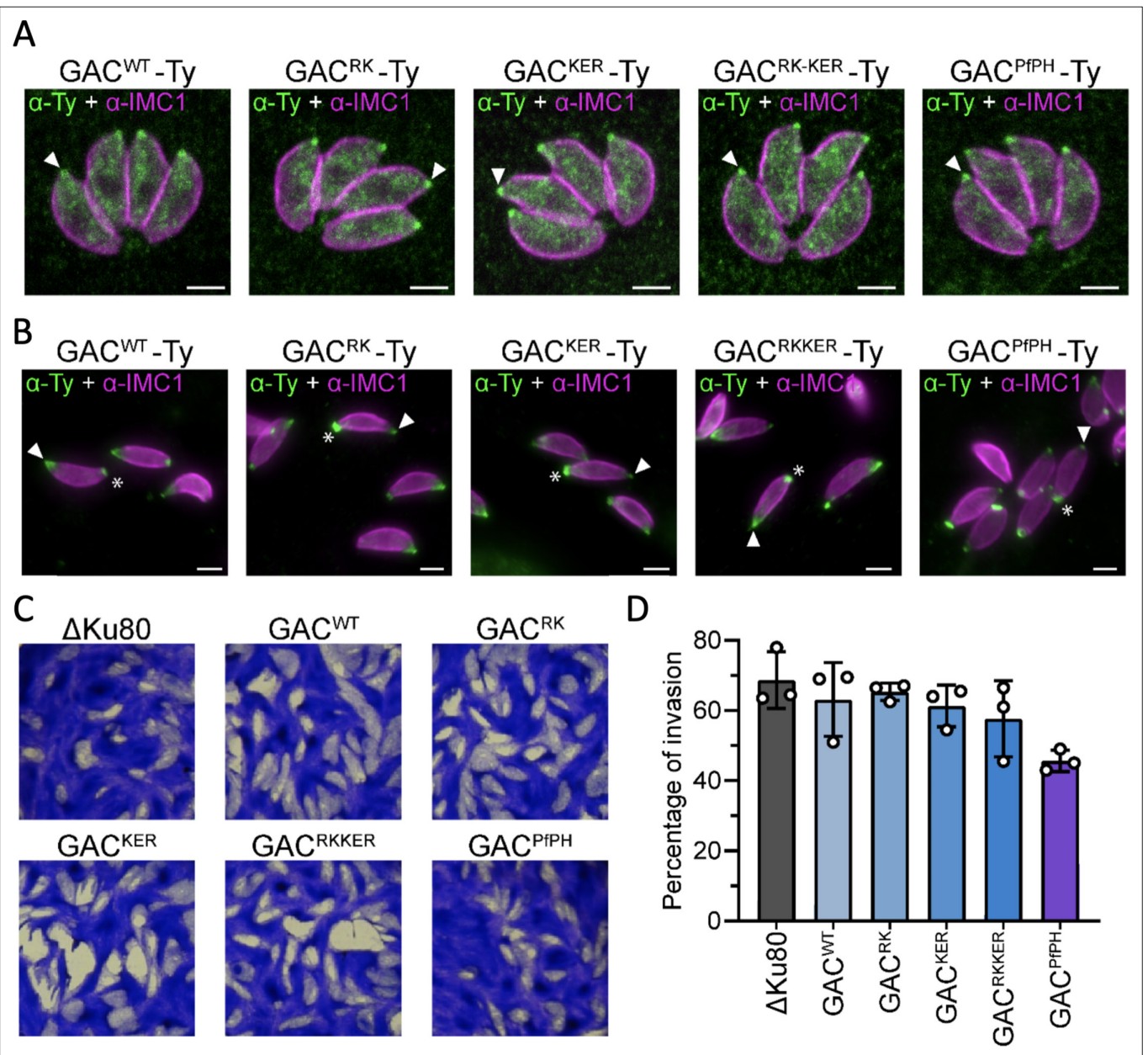

**Figure 6.** Phenotypical analysis of parasite bearing phosphatidic acid (PA)-binding mutations of glideosome-associated connector (GAC). (**A**) GAC localisation by Immunofluorescence assay (IFA) in intracellular parasites. White arrow = apical pole. Scale bar = 2 μm. (**B**) GAC localisation by IFA in extracellular parasites. White arrow = apical pole. White star = basal pole. Scale bar = 2 μm. (**C**) Plaque assay of the different mutants analysed. (**D**) Red/green invasion assay (n=3). A Δku80 background *T. gondii* strain is used, which is deficient in nonhomologous end joining and provides highly efficient gene replacement.

The online version of this article includes the following source data for figure 6:

**Source data 1.** Parasite invasion data tables.

and paired in quintuple mutant $RK_{2587–88}AA/KER_{2619–2621}AAA$. Interestingly, the $KER_{2619–2621}AAA$ had the largest effect on binding for the isolated PH domain, suggesting that other regions within full-length GAC may play a role in PA-enriched membrane binding.

## Key lipid-binding residues in the GAC PH domain are not essential for parasite invasion

*T. gondii* mutant lines were generated in which the PH domain was replaced at the endogenous locus with equivalent sequences containing the PA-binding mutations together with C-terminal Ty tag. Parasite lines were obtained with a wild-type PH domain (TgGACWT -Ty), with the individual targeted mutations (TgGAC$^{RK}$ -Ty and TgGAC$^{KER}$ -Ty), the double mutation patch (GAC$^{RKKER}$ -Ty), and a line in which the PH domain was replaced by the *P. falciparum* version (TgGAC$^{PfPH}$ -Ty). All GAC versions, wild-type, mutated and chimeric, localised correctly at the PCRs of the parasite as well as cytosolically (*Figure 6A*), suggesting that the PA-binding by the GAC PH domain is not critical for GACs apical localisation. Likewise, upon triggering parasite motility with 5-Benzyl-3-isopropyl-1H-pyrazolo [4,3-d]pyrimidin-7(6H)-one (BIPPO), all the strains displayed the characteristic basal accumulation of TgGAC (*Figure 6B*). Finally, to access if there was any fitness cost induced by the mutations, wild-type, and mutated parasites were analysed by plaque assay and revealed that the mutations were not detrimental for the parasite lytic cycle (*Figure 6C*). Accordingly, no clear defect in invasion could be observed in parasites expressing the mutated versions (*Figure 6D*). A small reduction of the plaque size and reduction in invasion capacity was observed for the TgGAC-Pf$_{PH}$ chimera.

## Evaluation of GAC and GAC fragments binding to *Toxoplasma* F-actin

Association to rabbit F-actin was previously shown to involve the N-terminal 1114 amino acids of TgGAC, roughly corresponding to the first 25 AHRs of the RI region (*Figure 7A and B*). To map further the interaction of TgGAC with the shorter and highly dynamic actin filaments characteristic of *T. gondii* (*Skillman et al., 2013*), a series of recombinant TgGAC fragments were purified and tested for binding (*Figure 7C*). Both full-length TgGAC and a fragment encompassing residues GAC$_{1–1114}$ co-sedimented with *Toxoplasma* F-actin produced from recombinant TgACT (*Figure 7D and E*). In contrast, a shorter N-terminal fragment, (TgGAC$_{1–619}$) corresponding to the first 14 AHRs, failed to interact with F-actin. This suggests that either the binding site for F-actin lies between residues 619 and 1114 of TgGAC or that that isolated N-terminal fragment TgGAC$_{1–619}$ adopts a different conformation than TgGAC$_{1–1114}$ and TgGAC which prevents its interaction with F-actin.

## Investigation of GAC fragment conformations using HDX-MS

To gain a deeper understanding into the conformational requirements of the binding of GAC to F-actin, we used HDX-MS. HDX-MS is a powerful method that examines protein dynamics by monitoring the exchange rate of protein amide hydrogens with the solvent (*James et al., 2022*). Information on protein secondary structure and conformation is provided by measuring overall deuteration levels, and the comparison of H/D exchange rates between protein constructs reveals differences in protein dynamics and conformation. To evaluate whether the 1–619 region adopts the same conformation in TgGAC$_{1–619}$, TgGAC$_{1–1114}$, and TgGAC-FL, HDX-MS analyses of peptides encompassing residues 1–619 were compared for all three constructs (*Figure 7F–I* and *Figure 7—figure supplement 1*; *Supplementary file 1*). Most of the GAC N-terminus exhibited identical dynamics in all constructs (*Figure 7F and G*), and only the C-terminal extremity of TgGAC$_{1–619}$ (residues 559–609) displayed higher H/D exchange rate compared to TgGAC$_{1–1114}$ and TgGAC-FL (*Figure 7F–I*). TgGAC$_{1–1114}$ and TgGAC-FL share identical dynamics within this region. This indicates that the two truncated fragments share a similar conformation as TgGAC-FL for residues 1–558, corresponding to a major portion (14 AHRs) of the first turn in the superhelical pyramid structure (19 AHRs). NMR data (*Figure 7—figure supplement 1C*) also confirms that this fragment is predominantly well folded. The C-terminal extremity of the shortest fragment, TgGAC$_{1–619}$, is significantly more flexible which also may contribute to its inability to bind F-actin.

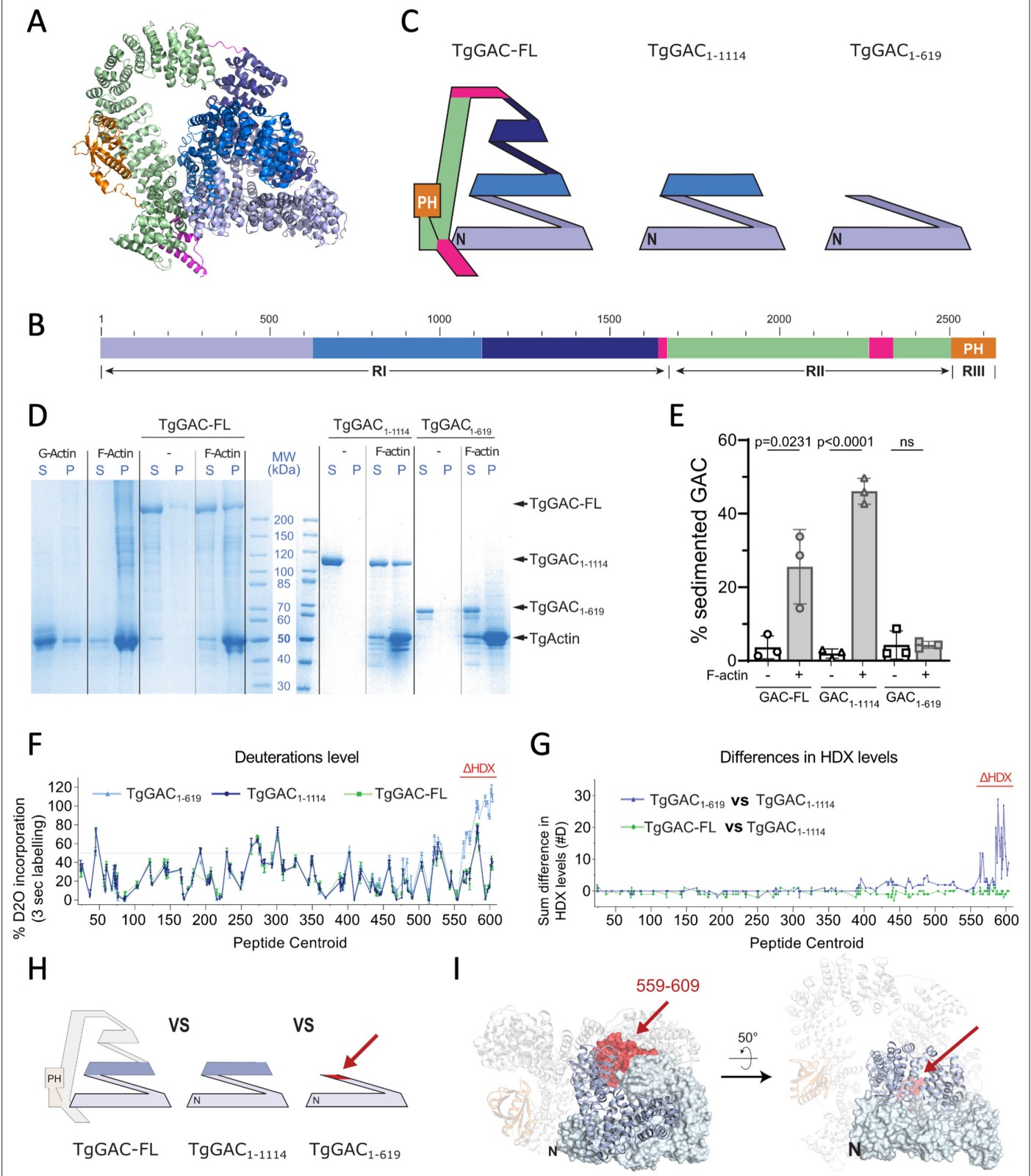

**Figure 7.** TgGAC interaction with TgActin. (**A**) Cartoon representation of the structure of TgGAC coloured according to schematic representation shown in (**B**). (**C**) Schematic representation of TgGAC structure coloured according to panel (**B**) with illustration of the truncated recombinant TgGAC constructs. (**D**) Coomassie-stained gel analysing proteins remaining in supernatant or sedimenting upon centrifugation at 100,000 g. S: Supernatant. P: Pellet. (**E**) Graphical representation of the percentage of TgGAC co-sedimenting alone or in the presence of TgActin filaments. Data are mean +

*Figure 7 continued on next page*

*Figure 7 continued*

SD of three independent experiments. p Value calculated using unpaired t-test. (**F**) Graph displaying deuteration levels for TgGAC$_{1-619}$, TgGAC$_{1-1114}$, and TgGAC-FL. Each dot represents a single peptide where the deuteration level in percentage maximal deuteration is plotted according to the residue number at the centre of the peptide (peptide centroid). (**G**) Comparison of H/D exchange levels for peptides spanning glideosome-associated connector (GAC) residues 1–619 between (i) TgGAC$_{1-619}$ and TgGAC$_{1-1114}$ (blue dots) and between (ii) TgGAC-FL and TgGAC$_{1-1114}$ (green). Each dot represents a peptide with the x-axis indicating peptide centroid as in (**F**). The y-axis indicates the sum of differences in number of deuterons incorporated for each peptide, measured at three deuteration times: 3, 30, and 300 s. Region showing significant differences between TgGAC$_{1-619}$ and TgGAC$_{1-1114}$ is indicated in red. (**H**) Schematic representation of the three GAC constructs studied by hydrogen/deuterium exchange coupled to mass spectrometry (HDX-MS), focussing on the region coloured in light blue (residues 1–619). Red colouring indicates the region where a difference in H/D exchange rate is observed. (**I**) Structure of TgGAC with fragment 1–619 shown as surface representation and highlighted in light blue. Residues 620–1114 are shown in blue, and the remaining of GAC structure is shown in light white. Region displaying a different conformation in TgGAC$_{1-619}$ compared to other TgGAC constructs, with increased H/D exchange rate, is coloured in red.

The online version of this article includes the following source data and figure supplement(s) for figure 7:

**Source data 1.** Full, uncropped SDS-PAGE images for F-actin binding experiments.

**Source data 2.** Figure and caption for uncropped gels of F-actin binding experiments.

**Figure supplement 1.** Hydrogen/deuterium exchange coupled to mass spectrometry (HDX-MS) and nuclear magnetic resonance (NMR) data for TgGAC$_{1-619}$ construct studied.

## Discussion

Apicomplexan parasites, for which *T. gondii* is a model organism, propel themselves by a specialised actomyosin-dependent gliding motility that relies on a large, conserved protein to connect actin filaments with parasite plasma membrane. GAC, an abundant protein that crucially localises to the PCRs that belong to the conoid complex at the apical tip of parasite, was shown to adopt multiple conformations to regulate its function (*Figure 1A*). The most striking feature of the TgGAC structure is the large continuous supercoiled ARM region (residues 1–2490) which forms a ring. The closure of the ring results from an extensive interface between the ARM1 and ARMs14-16 from RI and ARMs46-48 from RII (N/C-interface - *Figure 2*). This closed conformation was unexpected as an earlier SAXS model suggested an extended, club-shaped molecule in solution with no evidence for a N-/C-terminal interface (*Jacot et al., 2016*). Despite this discrepancy, the interface formed in the closed structure is highly conserved at the protein sequence level suggesting that it is a functionally relevant state. Subsequent pH-dependent SAXS analyses reported in the present study revealed that an open structure exists at high pH, but oligomerised/aggregrated forms predominate at low pH. The nature of the N/C interface in the closed structure is hydrophilic with several electrostatic interactions, including some that are readily titratable, such H24 and H2164. It is conceivable that deprotonation of these residues at high pH removes key salt bridges that stabilise the closed conformation and facilitate a conformationally labile open structure. Our SAXS data at low pH may provide some support for this notion, as it reveals a multimerised/aggregated state. While this is unlikely to reflect oligomerisation in vivo, it could result from the reformation of these electrostatic interactions in an intermolecular manner. The ability of GAC to adopt both a flexible open conformation and transition to a closed ring structure could play a role in regulating the multiple interactions of GAC assembly on the glideosome and to the parasite membrane. This behaviour is reminiscent of the unrelated ARM-rich protein, HUWE1, which is a quality-control E3 ligase, which forms a large, closed ring from 34 tandem ARM repeats. The closed structure is believed to represent a latent state but can adopt a range of open conformations to regulate its interactions with accessory proteins (*Grabarczyk et al., 2021*).

GAC interacts with PA-enriched membranes that are generated through a signalling cascade in activated parasites (*Bullen et al., 2016*). The C-terminal PH domain within GAC presents a conserved (*Figure 5* and *Figure 1—figure supplement 3*), positively charged surface, and while this is important for PA binding in vitro, its mutation does not cause a significant defect in GAC localisation, translocation, or parasite fitness. Other regions in GAC, such as the basic protrusion in AHR53 (*Figure 3* and *Figure 1—figure supplement 3*), are also likely to contribute to membrane binding in vivo. Furthermore, the presence of a membrane interaction surface within the TgGAC$_{PH}$ suggests that the binding site for the juxtamembrane region of the MIC2 cytoplasmic tail is located nearby, and these interactions may also be cooperative. Even though the PH domain represents a minor portion of the full-length GAC structure, its interaction with RII region seems crucial for GAC structural integrity, as a construct lacking the PH domain cannot fold properly.

The principal *Toxoplasma* F-actin binding region is localised to the first 1114 residues of TgGAC (TgGAC$_{1-1114}$), which forms a large, supercoiled base of the N-terminal pyramid (TgGAC-R1:1–1656 – *Figure 2*) and provides a platform with several potential sites of interaction with a helical actin filament. A shorter fragment encompassing only the first turn of the supercoiled pyramid (TgGAC$_{1-619}$) does not interact with F-actin. Analysis of the protein conformation by HDX-MS showed that TgGAC$_{1-619}$ adopts a similar conformation for residues 1–558 as in full-length TgGAC, suggesting that the minimal actin-binding interface lies between residues 559–1114 (AHRs 15–25) in which the second supercoiled turn starts. The absence of F-actin binding for the N-terminal portion (1–619), which forms the base of GAC pyramidal structure, suggests a role in this region in stabilising the closed conformation as it makes direct contact with the C-terminal end of the RII region. Membrane association simulations for TgGAC reveal a specific membrane-binding surface involving the PH domain and RII, and this orientation places the TgGAC$_{1-1114}$ region distal for interaction with the actin filament.

As the available pellicular space between the parasite inner membrane complex and the plasma membrane is insufficient for the open structure of GAC (i.e. with RII and RIII extended) to bridge F-actin to the plasma membrane lengthways, the closed structure is likely to represent a functional important state of cytosolic GAC or during GACs initial engagement within the glideosome. Other actin-membrane bridging proteins, like the ERM (ezrin, moesin, and radixin) family of proteins, cross-link cortical actin to plasma membrane, and full engagement is achieved by a reorganisation of actin-binding regions by cooperative interactions with phosphatidylinositol 4,5-bisphosphate (PIP2; *Ben-Aissa et al., 2012*). It is therefore conceivable that PA binding and MIC2 recruitment at the plasma membrane by GAC contribute to conformational changes to GAC structure, i.e., a transition from closed to open and extended structures.

The function of GAC in bridging parasite cytoskeleton to the host cell substrate is reminiscent of that for the mammalian catenins within adherens junctions, which comprise several components. The C-terminal AHR arch region (AHRs 38–53) displays significant structural similarity with the armadillo repeat region (ARM) of the ß-catenins, superimposing with an RMSD of 4.2 Å over the backbone of 348 equivalent residues (*Figure 1—figure supplement 4*). Nominally, α-catenin crosslinks F-actin to β-catenin, while β-catenin establishes the connection to the E-cadherin tails. The single chain of GAC carries out both these roles, i.e., RI acts like the actin-binding α-catenin and is linked via L1 to RII ARM arch structure, which like β-catenin tethers the system to the cell surface adhesins. The ß-catenins bind extended cadherin tails emerging from the plasma membrane via a superhelical surface formed by 12 tandem ARM repeats (*Choi et al., 2009*; *Huber and Weis, 2001*; *Ishiyama et al., 2010*). As TgGAC likely binds the cytoplasmic C-terminal tails of the plasma membrane adhesin TgMIC2 (*Jacot et al., 2016*), it is tempting to speculate that a similar interface formed in ß-catenin/E-cadherin complexes is used by TgGAC to link TgMIC2 to the glideosome machinery (*Figure 1—figure supplement 4*).

Intriguingly, the extensive interface within and between the catenins and E-cadherin facilitates molecular transduction via mechanical strain (*Angulo-Urarte et al., 2020*; *Bush et al., 2019*; *Valenta et al., 2012*). Mechanical force induces a rearrangement of binding interfaces that result in a strengthening of its interaction with F-actin, and this effect is dependent on the direction of applied force (*Mei et al., 2020*; *Xu et al., 2020*). Such catch bond behaviour may also be relevant for GAC function. Importantly, part of the 559–1114 F-actin binding region is not fully accessible in the closed conformation; therefore, it is also conceivable that optimal F-actin binding requires significant structural rearrangement of the closed structure. The stabilised closed conformation would be able to both resist and sense the significant inter parasite–substrate forces generated by MyoA translocation of F-actin. Such force acting along parasite–host cell interface could induce opening of the GAC structure and unveil additional cryptic binding sites that strengthen F-actin binding and ensure a coordinated direction of motion. The lower affinity of full-length TgGAC compared to TgGAC$_{1-1114}$ for F-actin supports this hypothesis (*Figure 7E*).

Structural similarity also exists between the C-terminal ARM region of GAC (TgGAC$_{1670-2639}$) and the family of myosin-specific chaperones which possess ARM-rich UCS (UNC-45/Cro1/She4) domains (*Hellerschmied and Clausen, 2014*) that interact with myosin motors domains (*Figure 1—figure supplement 4*). While *T. gondii* possesses a dedicated UCS chaperone for TgMyoA (TgUNC; *Bookwalter et al., 2014*), which is critical for successful folding of all the parasite myosin motors (*Frénal et al., 2017b*), the similarity between the GAC C-terminal ARM region and the UCS chaperones is intriguing. Myosin chaperones also have the propensity to multimerise and, in some cases, form chains

that assist myosin assembly on the filament (*Gazda et al., 2013*). Although no direct evidence for an interaction between TgGAC and TgMyoA has been found, it is tempting to speculate that TgGAC may assist TgMyoA organisation on F-actin.

# Materials and methods

## Protein expression and purification

Full-length TgGAC gene with TEV cleavable N-terminal 6xHis-tag has been cloned into the pET28a vector as previously described (*Jacot et al., 2016*; *Kumar et al., 2022*). Constructs for PfGAC$_{PH}$ (PfGAC$_{2471–2605}$) and TgGAC$_{PH}$ (TgGAC$_{2505–2639}$) were constructed with a 6His purification tag and an additional SUMO tag for soluble expression of PfGAC$_{PH}$. A Q5 site-directed mutagenesis kit (NEB) and the manufacturers protocol were used to generate and PfGAC$_{PH}$ mutants with standard primers. For protein expression, plasmids were transformed into BL21 (DE3; NEB) or Rosetta2 (Novagen) *Escherichia coli* strains. Expression was carried in minimal medium supplemented with $^{15}$NH4Cl and/or $^{13}$C-glucose for NMR isotopic labelling. Purification for His-tagged constructs was carried by Ni2+ affinity chromatography. Removal of the SUMO tag for PfGAC$_{PH}$ samples was carried out by incubation with purified ULP1 protease. Further purification for all constructs was achieved by size-exclusion chromatography.

## NMR spectroscopy

Samples of purified $^{15}$N/$^{13}$C-PfGAC$_{PH}$ and $^{15}$N/$^{13}$C-TgGAC$_{PH}$ were prepared and supplemented with D$_2$O. All NMR spectra were acquired at 298 K on Bruker Avance-III DRX 800 and Avance-III 600 spectrometers. Triple resonance HNCA, HNCACB, HNCO, and HN(CO)CA spectra (*Sattler et al., 1999*) were recorded and analysed to obtain backbone assignments, which was assisted using MARS program (*Jung and Zweckstetter, 2004*). Chemical shift assignment and analysis were performed using an in-house version of NMRview (*Marchant et al., 2008*). HBHA(CO)NH, H(CCO)NH, CC(CO)NH, and HCCH-TOCSY spectra were recorded for use in side-chain chemical shift assignment. $^{15}$N-NOESY and $^{13}$C-NOESY spectra were recorded and used as distance restraints in structural calculation/validation.

For PfGAC$_{PH}$ and TgGAC$_{PH}$ 1D $^1$H NMR LUV binding assays performed with LUVs containing an increasing proportion of POPA (POPA Mol% value). To form LUVs, a lipid suspension was sonicated until transparent and then clarified by centrifugation. The supernatant containing LUVs were prepared at an 8 mM total lipid concentration for use in 1D NMR LUV-binding assays. Dynamic light scattering analyses show homogeneous LUV diameters of ~100 nm. For each LUV composition titration series, a separate $Kd_{app}$ value was calculated for each replicate (n=3), and a mean $Kd_{app}$ value was calculated. PfGAC$_{PH}$ and TgGAC$_{PH}$ titration PRE experiments were carried out with increasingly POPA-enriched MSP1D14-5 nanodiscs. MSP1D1H4-5 MSP was expressed and purified by nickel affinity chromatography using a well-established protocol (*Ritchie et al., 2009*). Peak elution fractions containing pure nanodiscs (assessed by SDS-PAGE) from size-exclusion chromatography were pooled, concentrated, and stored (at –80°C). Relative NMR signal reductions for amide resonances in $^1$H-$^{15}$N HSQC spectra were determined and mapped. For 1D 1H-NMR LUV binding assays, the fraction of bound protein at each titration point was averaged between replicates and plotted as the mean. Error bars for binding curves represent 1σ from this mean value. Error bars shown for $Kd_{app}$ values represent 1σ for mean $Kd_{app}$ from fitting binding isotherms.

## X-ray data collection and processing

Diffraction data from a single native crystal were collected on beamline i04 of the Diamond Light Source (DLS), UK. Data were processed with CCP4 dials (*Beilsten-Edmands et al., 2020*; *Winn et al., 2011*; *Winter, 2010*; *Winter et al., 2018*) and scaled using dials.scale (*Evans, 2006*) within the Xia2 package (*Winter et al., 2013*). Multi-wavelength anomalous diffraction (MAD) data from a single SeMet labelled crystal were collected on beamline i04 of the DLS at the following wavelengths: peak = 0.9795 Å, inflection = 0.9796 Å, and remote = 0.9722 Å. Data were processed initially by Auto-Proc (*Vonrhein et al., 2011*). Substructure definition and initial model building were performed using AutoSHARP (*Vonrhein et al., 2007*). This was followed by manual building in Coot (*Emsley et al., 2010*) and further refinement using Phenix Refine (*Adams et al., 2010*). Data collection statistics have

been published previously (**Kumar et al., 2022**). The structure has been deposited in wwPDB under accession code: PDB ID 8C4A (**Berman et al., 2007**).

## Small angle X-ray scattering

Samples for SAXS measurements were prepared by concentrating samples from SEC using a centrifugal spin device with a molecular weight cut-off of 100 kDa. For all experiments, a buffer of 25 mM Tris, 5 mM TCEP, and pH 8.0 was used, which was adjusted to a pH value between 4.0 and 8.0 by addition of 1 M HCl leading to a salt concentration of maximum 30 mM in the case of pH 4.0. SAXS data were measured using a laboratory based flux-optimised Bruker AXS Nanostar with a gallium liquid metal jet source (**Schwamberger et al., 2015**) and scatterless slits (**Li et al., 2008**), a detector distance of 867 mm, and a $q$ range of 0.0098–0.42 Å$^{-1}$. The entire $q$ range could be probed with this setup in a single measurement. More information about the optimised instrument can be found here (**Lyngsø and Pedersen, 2021**). All data were measured for 1800s at 20°C. SAXS data are plotted as an intensity as a function of $q$, which is the modulus of the scattering vector and defined as $q = (4\pi\sin[\Theta])/\lambda_{Ga}$, where $2\Theta$ is the scattering angle between the incident and scattered beam and $\lambda_{Ga} = 1.34$ Å. Data were background subtracted and converted to absolute scale using the software package SUPER-SAXS (CLP Oliveira and JS Pedersen). The mass was calculated using $M = \left(I\left(0\right) \times N_A\right) / \left(c \times \Delta\rho_m^2\right)$, where $I(0)$ is the intensity extrapolated to $q$=0, $N_A$ is Avogadro's number, $c$ is the protein concentration in g/mL, and $\Delta\rho_m$ is the scattering contrast per mass that can be estimated to $2.0 \times 10^{10}$ cm/g for a typical protein. $I(0)$ was determined both by a Guinier fit analysis (using the intercept with the y-axis) and from an IFT routine (**Glatter, 1977**). The theoretical scattering curve was calculated using the program wlsq_PDBx (**Steiner et al., 2018**). An ab initio reconstruction of the protein structure was performed using GASBOR from the ATSAS package (**Svergun et al., 2001**) where the number of amino acids is given, and each amino acid is represented by a dummy residue. The optimisation was performed with the real space option due to the large size of GAC.

## Molecular dynamics simulations

The TgGAC crystal structure was used to generate coarse-grained protein:membrane simulation systems in which TgGAC was rotated randomly in respect to the membrane patch to generate the first set of coordinates. CHARMM-GUI Martini Maker was used to generate a system in which GAC was translated 12 nm in the z direction from the centre of the membrane (**Qi et al., 2015**). The membrane composition used was 50% POPC: 50% POPA to replicate simulations performed in **Darvill et al., 2018**. For two independent replicates, TgGAC was rotated again (0, 90) and (90, 0). Simulations were performed using the gromacs biomolecular software package version 2021.3 (**Hess et al., 2008**) and the MARTINI3 forcefield with ElNeDyn restraints (**Souza et al., 2021**). The v-rescale thermostat (tau 1.0 ps; **Bussi et al., 2007**) and the Parrinello–Rahman barostat (tau 12.0 ps; **Parrinello and Rahman, 1981**) were used to maintain temperature (303.15 K) and pressure (1 bar). Production simulations were 10 μs in length. Analysis was performed using gromacs tools, VMD (**Humphrey et al., 1996**), and the ProLint server (**Sejdiu and Tieleman, 2021**).

To generate a model for a TgGAC open conformation, we used steered-MD. The TgGAC crystal was first used to generate a coarse-grained model, using the CHARMM-GUI Martini Maker (**Qi et al., 2015**) with the MARTINI3 forcefield and the ElNeDyn restraints (**Souza et al., 2021**). The restraint network at the N-/C-terminal interface was removed using an in-house Python script. A small and optimised pulling force was applied (pull rate = 0.01 nm/ns, k=100 kJ/mol.nm$^2$) between a selected helix on either side of the interface to drive its separation and the extension of the structure. A series of structures along the trajectory were generated over the course of an averaged 2.5 μs simulations in length. The procedures were repeated to drive the separation of the three supercoiled pyramid interfaces (TgGAC$_{1-619}$, TgGAC$_{620-1114}$, and TgGAC$_{1114-1661}$). The snapshots were back mapped to atomistic models using Martini to All-atom Converter (**Wassenaar et al., 2014**) and used in the fits of the SAXS data.

## Plaque assay

Human foreskin fibroblasts (HFFs) monolayers were infected with freshly egressed parasites and incubated for 7 days at 37°C. Cells were then fixed with 4% paraformaldehyde (PFA)/0.05% glutaraldehyde

for 10–15 min. After neutralisation with 0.1 M glycine/PBS, cells were stained using crystal violet. For quantification, pictures were taken, and the plaque area was determined using ImageJ.

## Immunofluorescence assay

For vacuoles images, parasites were inoculated on an HFF monolayers previously seeded on a glass coverslip. The parasites were grown for 16–24 hr at 37°C. For extracellular parasites, freshly egressed parasites were seeded on gelatin-coated coverslips, and media containing BIPPO was used to stimulate motility. The coverslips were then fixed with 4% PFA/0.5% glutaraldehyde (PFA-Glu) during 10 min. The fixative agent was then neutralised using 0.1 M glycine/PBS for 10 min. Cells were then permeabilised for 20 min using 0.2% TritonX100/PBS and blocked using 5% BSA/PBS for 20 min before incubation with primary antibodies in 2% BSA/0.2% TritonX100/PBS for 1 hr. Coverslips were then washed three times for 5 min using 0.2% TritonX100/PBS. Secondary antibodies were incubated for 1 hr similarly to the primary antibodies. Finally, the coverslips were washed three times with PBS before mounting on glass slides using DAPI-containing Fluoromount. For immunofluorescence analysis, the secondary antibodies Alexa Fluor 488 and Alexa Fluor 594 conjugated goat α-mouse/rabbit antibodies (Molecular Probes) were used.

## Liposome binding assay

Proteins were expressed in bacteria and purified by the PPR2P platform (University of Geneva). The liposomes were prepared in-house as follow: lipids were mixed in glass vials with the following proportions: 10% DOPE +X%DO-PA + qsp%DOPC (percentage by weight). The mix was slowly dried using nitrogen flow. The lipids were dried further in a dessicator for 30 min. The dried lipids were then resuspended in 'lipid buffer' (50 mM Hepes pH 7.5/100 mM NaCl/5% glycerol) to reach a 5 mg/mL concentration. Resuspension was then ensured by vortexing the mixes for 5 min. Then, seven freeze–thaw cycles were performed (20 s in liquid nitrogen followed by 90 s in a 33°C water bath). Lipid mixes were extruded by passing them through 0.1 μm filters for 21 times. Liposomes were then aliquoted in microtubes (50 μL), flash frozen in liquid nitrogen, and stored at –80°C until use.

For the binding assay, proteins were resuspended in 'protein buffer' (20 mM Tris pH 7.4/200 mM NaCl/1 mM DTT), centrifuged at 100,000 g for 30 min at 4°C to remove any precipitate and measure the exact protein concentration in the supernatant. Protein concentration was then adjusted to 0.2 mg/mL using protein buffer. In parallel, the liposomes were diluted at 2 mg/mL in 'lipid buffer' (50 mM Hepes pH 7.4/100 mM NaCl/1% glycerol) and re-diluted twofold using water to achieve a concentration of 1 mg/mL. For the 'no lipid' conditions, lipid buffer was simply diluted twofold in water. For the reaction, 40 μL of protein and 40 μL of liposomes were mixed and incubated on ice for 1 hr. The mix was then centrifuged at 120,000 g for 30 min at 4°C. Supernatant and pellets were separated, resuspended in Laemmli buffer, and boiled 10 min at 95°C. Equal volumes were then ran on polyacrylamide gels and stained using Coomassie blue. Quantification was performed by band densitometry.

## HDX-MS sample preparation and data analysis

HDX-MS experiments were performed at the UniGe Protein Platform (University of Geneva, Switzerland) following a well-established protocol with minimal modifications (**Wang et al., 2018**). Details of reaction conditions and all data are presented in **Supplementary file 2** and **Supplementary file 3**. HDX reactions were done in 50 μL volumes with a final protein concentration of 2.4 μM of GAC protein. Briefly, 120 pmol of the protein in 10 μL final volume were pre-incubated 5 min at 22°C before the reaction.

Deuterium exchange reaction was initiated by adding 40 μL of $D_2O$ exchange buffer (20 mM Tris pH 8, 150 mM NaCl, and 5 mM DTT in $D_2O$) to the protein–peptide mixture. Reactions were carried-out at room temperature for three incubation times (3 s, 30 s, and 300 s) and terminated by the sequential addition of 20 μL of ice-cold quench buffer 1 (4 M Gdn-HCl, 1 M NaCl, 0.1 M $NaH_2PO4$ pH 2.5, 1% formic acid [FA], and 200 mM TCEP). Samples were immediately frozen in liquid nitrogen and stored at –80°C for up to 2 weeks. All experiments were repeated in triplicate.

To quantify deuterium uptake into the protein, samples were thawed and injected in a ultra-performance liquid chromatography (UPLC) system immersed in ice with 0.1% FA as liquid phase. The protein was digested via two immobilised pepsin columns (Thermo #23131), and peptides were collected onto a VanGuard precolumn trap (Waters). The trap was subsequently eluted, and peptides

were separated with a C18, 300 Å, 1.7 µm particle size Fortis Bio 100×2.1 mm column over a gradient of 8–30% buffer C over 20 min at 150 µL/min (buffer B: 0.1% FA; buffer C: 100% acetonitrile). Mass spectra were acquired on an Orbitrap Velos Pro (Thermo), for ions from 400 to 2200 *m/z* using an electrospray ionisation source operated at 300°C, 5 kV of ion spray voltage. Peptides were identified by data-dependent acquisition of a non-deuterated sample after MS/MS, and data were analysed by Mascot. All peptides analysed are shown in *Supplementary file 2* and *Supplementary file 3*. Deuterium incorporation levels were quantified using HD examiner software (Sierra Analytics), and quality of every peptide was checked manually. Results are presented as percentage of maximal deuteration compared to theoretical maximal deuteration. Changes in deuteration level between two states were considered significant if >20% and >2 Da and p<0.01 (unpaired t-test).

## Strains generation – GAC mutations at the endogenous locus

The pGST-GAC_PH domain vector used to produce the PH domain of GAC was used as a base to generate the points mutation. The points mutations were inserted using the Q5 mutagenesis kit (New England Biolabs) and specific primers. The mutations in the PH domains were checked by sequencing. Then, the 5′UTR_UPRT-pT8-DDGACΔPH-Ty-DHFR-3′UTR_UPRT was linearised with SgrAI, while the mutated PH domain from the pGST vector was amplified by PCR using primers containing homology regions. The mutated PH domain was inserted in the linearised 5′UTR_UPRT-pT8-DDGACΔPH-Ty-DHFR-3′UTR_UPRT by Gibson assembly. The insertion of the PH domain in the receiving vector was checked by analytical digestion and sequencing. The mutated PH domain of newly generated 5′UTR_UPRT-pT8-DDGACmut-Ty-DHFR-3′UTR_UPRT vector (therefore without introns) was amplified by KOD PCR using specific primers containing homology regions (on one side with the sequence preceding the PH domain, and on the other with the 3′UTR of the GAC gene). A double gRNA targeting the region preceding the PH domain and the 3′UTR of the GAC gene was generated in parallel by Q5 mutagenesis. For transfection, the equivalent of 100 µL of KOD PCR and 40 µg of gRNA was used. 48 hr after transfection, the parasites were fluorescence-activated cell sorted (FACS) to select clones expressing the gRNA-Cas9-YFP construct. Then, the individual clones were amplified, genomic DNA was extracted, and integration PCR of the GAC PH domain region was checked (the PCR ensured the loss of the introns, generating a smaller PCR amplicon in modified parasites compared to wild types). In addition, the PH domain of the PCR positive clones was fully sequenced. Primers are shown in *Supplementary file 4*.

## Materials and data availability

Reagents generated in this study will be made available upon request. All data generated or analysed during this study are included in the manuscript and supplementary files. DNA primers are listed in *Supplementary file 4*. The structure has been deposited in wwPDB under accession code: PDB ID 8C4A (*Berman et al., 2007*). The mass spectrometry proteomics data have been deposited to the ProteomeXchange Consortium via the PRIDE (*Perez-Riverol et al., 2022*) partner repository with the dataset identifier PXD039335.

## Acknowledgements

This work was supported by a BBSRC and Leverhulme Trust awards to SJM (BB/W001764/1 and RPG_2018_107), the Independent Research Fund Denmark through grants 8021-00133B (JSP), and the Swiss National Science Foundation to DSF (10030_185325 and CRSII5_198545). The crystallisation facility at Imperial College London is supported by the Biotechnology and Biological Sciences Research Council (BB/D524840/1) and Wellcome Trust (202926/Z/16/Z). We are grateful to staff at Diamond Light Source beamline I04 for their help with data collection. We are thankful to Rémy Visentin at the Protein Platform of the University of Geneva for assistance with recombinant protein purification.

## Additional information

### Competing interests
Dominique Soldati-Favre: Senior editor, eLife. The other authors declare that no competing interests exist.

### Funding

| Funder | Grant reference number | Author |
|---|---|---|
| Leverhulme Trust | RPG_2018_107 | Stephen Matthews |
| Biotechnology and Biological Sciences Research Council | BB/W001764/1 | Stephen Matthews |
| Swiss Re Foundation | 10030_185325 | Dominique Soldati-Favre |
| Swiss Re Foundation | CRSII5_198545 | Dominique Soldati-Favre |
| Biotechnology and Biological Sciences Research Council | BB/D524840/1 | Stephen Matthews |
| Wellcome Trust | 202926/Z/16/Z | Stephen Matthews |
| Independent Research Fund Denmark | 8021-00133B | Stephen Matthews |

The funders had no role in study design, data collection and interpretation, or the decision to submit the work for publication. For the purpose of Open Access, the authors have applied a CC BY public copyright license to any Author Accepted Manuscript version arising from this submission.

### Author contributions
Amit Kumar, Kin Chao, Jan Skov Pedersen, Formal analysis, Investigation, Methodology, Writing – review and editing; Oscar Vadas, Formal analysis, Investigation, Methodology, Writing – original draft, Writing – review and editing; Nicolas Dos Santos Pacheco, Formal analysis, Investigation, Visualization, Methodology, Writing – original draft, Writing – review and editing; Xu Zhang, Gloria Meng-Hsuan Lin, Fisentzos A Stylianou, Investigation, Methodology, Writing – review and editing; Nicolas Darvill, Yingqi Xu, Investigation, Visualization, Methodology, Writing – review and editing; Helena Ø Rasmussen, Sarah L Rouse, Marc L Morgan, Formal analysis, Investigation, Visualization, Methodology, Writing – review and editing; Dominique Soldati-Favre, Conceptualization, Formal analysis, Supervision, Funding acquisition, Investigation, Methodology, Writing – original draft, Project administration, Writing – review and editing; Stephen Matthews, Conceptualization, Formal analysis, Supervision, Funding acquisition, Investigation, Visualization, Methodology, Writing – original draft, Project administration, Writing – review and editing

### Author ORCIDs
Oscar Vadas http://orcid.org/0000-0003-3511-6479
Nicolas Dos Santos Pacheco http://orcid.org/0000-0003-1959-194X
Helena Ø Rasmussen http://orcid.org/0000-0001-8384-656X
Dominique Soldati-Favre http://orcid.org/0000-0003-4156-2109
Stephen Matthews http://orcid.org/0000-0003-0676-0927

### Decision letter and Author response
Decision letter https://doi.org/10.7554/eLife.86049.sa1
Author response https://doi.org/10.7554/eLife.86049.sa2

## Additional files

### Supplementary files
• Supplementary file 1. Table of deuterium incorporation for each of the selected peptides used for hydrogen/deuterium exchange coupled to mass spectrometry (HDX-MS) analysis are presented.

Results are shown both as percentage deuteration compared to a theoretical maximal deuteration level and as number of deuterons incorporated into the peptide (#D).

• Supplementary file 2. Table of differences in deuterium uptake between two states. For each state comparison, results are shown for both differences in deuterium uptake by percentage (%D) and by number of deuterons (#D). Largest differences are highlighted in different colours: blue indicates PROTECTION of the peptide, and orange indicates increased EXPOSURE (usually associated with allosteric modifications).

• Supplementary file 3. Table for hydrogen/deuterium exchange (HDX) experimental details.

• Supplementary file 4. Key DNA primers used generate mutations and the endogenous locus mutants for parasite experiments.

• MDAR checklist

## Data availability

Diffraction data have been deposited in PDB under the accession code 8C4A. The mass spectrometry proteomics data have been deposited to the ProteomeXchange Consortium via the PRIDE partner repository with the dataset identifier PXD039335. All data generated or analysed during this study are included in the manuscript, figures and supplementary files.

The following datasets were generated:

| Author(s) | Year | Dataset title | Dataset URL | Database and Identifier |
|---|---|---|---|---|
| Kumar A, Morgan RML, Matthews SJ | 2023 | Structural and regulatory insights into the glideosome-associated connector from *Toxoplasma gondii* | https://www.rcsb.org/structure/unreleased/8C4A | RCSB Protein Data Bank, 8C4A |
| Kumar A, Morgan RML, Matthews SJ | 2023 | Structural and regulatory insights into the glideosome-associated connector from *Toxoplasma gondii* | https://www.ebi.ac.uk/pride/archive/projects/PXD039335 | PRIDE, PXD039335 |

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
