## [Editor Report]

The authors describe the first full–length crystal structure and solution conformation of the glideosome–associated connector (GAC) protein from *Toxoplasma gondii*. The data are convincing and support a model in which GAC uses multiple conformations and lipid–binding surfaces. This study is an important step towards a mechanistic understanding of glideosome assembly and function during the invasion process.

---

## [Decision Letter]

**Decision letter after peer review:**

Thank you for submitting your article "Structural and regulatory insights into the glideosome– associated connector from *Toxoplasma gondii*" for consideration by *eLife*. Your article has been reviewed by 3 peer reviewers, and the evaluation has been overseen by a Reviewing Editor and Amy Andreotti as the Senior Editor. The reviewers have opted to remain anonymous.

Essential revisions:

1) Although the structure of GAC is overall well described it requires further polishing in terms of model quality and accuracy (as it stands it appears like a structure at an early stage of refinement). Please carefully address the reviewer comments as detailed below in the recommendations for the authors.

2) Alternative interpretation of the SAXS data should be considered concerning the oligomerization state of GAC induced by pH.

3) The manuscript should be carefully reviewed to correct numerous errors in the text, figures and legends.

4) The model presented in Figure 8 is too speculative and the figure should be removed.

*Reviewer #1 (Recommendations for the authors):*

– The crystal structure model a long way from being good enough in quality and appears like a structure at an early stage of refinement. It has a large number of Ramachandran and bond length and angle outliers. Looking at the model, there are also large areas in which the model does not fit the map. I have not mentioned this in the public report, to be kind, but this really must be fixed, or it will be embarrassing. A structure at this resolution should really have no outliers (except perhaps for a few Ramachandran outliers if the density shows them to be genuine) of these types and should fit the maps.

– Figure 1 does not make clear what is a model and what an experimentally derived structure is. I would show the NMR and crystal structures as the first two panels and only then the composite. In the era of AlphaFold, it is important to distinguish clearly between experimentally determined structures and models.

– There are quite a lot of errors in the text and figures and legends.

– Error in Figure 4 legend – two (e) and no (f).

– Error in Figure 5 legend – there is no (d) or (e) in the figure.

– Comment on the differential effect of mutations – for example, KER – on full–length and PH domains.

– Figure 7d/e – what are GAC N100 and GAC N70? What is peptide centroid? Why is this not shown for the full protein length when GAC fl has been studied in this way? What is e?

– It felt to me that the data in Figure 7d/e would fit better with the SAXS data and make more sense to readers here when describing the flexibility of GAC.

– Personally I would remove Figure 8 as it is extremely speculative.

*Reviewer #2 (Recommendations for the authors):*

1. Structure (X–ray and NMR) description.

The full–length and quite large (2639 residues) structure was solved in a composite way. The largest part (7–2504) solved Se–MAD crystallography revealing 53 modules resembling armadillo/heat–like repeats and a small (2505–2639) and dynamic pleckstrin homology–like domain C–terminal domain solved by NMR assisted with AlphaFold2 modelling.

The structure is well–described and illustrations are relatively clear to get a sense of the modular organization of this large modular protein. Figure S2 in particular is to be mentioned as the Authors took the time to generate a SS mapping of the entire large structure on the sequence.

This reviewer is a crystallographer and inspected the density map in its entirety in COOT using the PDB and two maps provided 2Fo–Fc and Fo–Fc (thank you for the PDB and maps). It is a large protein so I am aware it represents a considerable amount of work but the repetitive modular organization is actually helpful to guide building (especially with other related structures at hand). This structure still requires some significant polishing in some areas where things can be improved and clearly cleaned up especially for this resolution range considering it was phased by MAD.

A lot of seleno methionines need to be adjusted better in the maps. Especially the Selenium atoms are electron–dense so it should be easy to fix. Quite a few side chains are built in but have no density, might want to do something about that and a few missing could be built in or readjusted.

I listed down some of the most obvious problems.

SeMet 30

R84

SeMet 89

S120

Q125

E128

T160 <->Q203

Loop 275-285. Discontinuous there but strong residual density

Fragment 326-329 is problematic

Q403

E420

E437 backbone trace?

Y488

K525 and R560 no sidechain visible

E745

D787-V790 check the backbone. This is wrongly traced.

H810 area is weak. Bad geometry of the backbone around that turn. Please inspect.

Q957/Q959

Q981

K1041

T1069 rotamer?

R1166

E1183

D1221

K1321

SeMet 1408 is plain broken in my coot?

E1510

E1530

SeMet 1609

D1663

R1667

SeMet 1707

E1749

K1752

R1832

H1988

SeMet 2010

Segment L2052-T2056 is problematic

SeMet 2343 and 2349

E2412

SeMet 2416

I see a disulfide bond between residues C2420 and C2467 but that looks really wrong. It is a cytoplasmic protein in the parasite and was expressed in *E. coli*. The maps show that the C2420 SH and C2467 point outwards and would not be engaged into an S-S bridge that looks wrong in the Fo-Fc negative maps. So recheck this and refine accordingly. C2420 looks towards T2470 while C2467 would rather look towards the backbone carbonyl of T2466.

Would these cysteines be conserved in Plasmodium?

Recheck your waters. Some dense peaks could be assigned to waters.

The statistics for crystal structure resolution (i.e., Se-MAD phasing) are not reported accurately. This should be remedied.

Phasing and refinement.

Data quality and refinement/phasing.

Outer shell statistics with *Rmerge* and Rmeas of over 100% with an I/σ(I) of only 0.71 are a bit extreme even to me. I know that the contributions of weak reflections are weighted down with all the maximum likelihood refinement procedures but still. Please report the multiplicity/redundancy of the overall data set and the outer shell in this present version of the study.

Table 1. Since no phasing information is reported there, it would be adequate to remove phasing from its title. It is only data quality and refinement statistics.

That said it is expected to scientifically and rigorously report the phasing statistics since no phasing statistics can be found anywhere in this manuscript. I am sure the first and last authors will agree with me on this. Citing the other article *, which however does report multiplicity, is of no interest in this case.

Kumar A, Zhang X, Vadas O, Stylianou FA, Dos Santos Pacheco N, Rouse SL, Morgan ML, Soldati–Favre D, Matthews S (2022) Secondary Structure and X–rayCrystallographic Analysis of the Glideosome–Associated Connector (GAC) from *Toxoplasma gondii*. Crystals 12

Refinement. In light of what I saw while inspecting the whole structure and ED Fourier difference maps, it might be useful to consider anisotropic TLS refinement of atomic displacement parameters, especially in light of the 'subdomain' structures suggested by SAXS/HDX and MD simulation data. Coloring the structure by the B factor on the backbone is quite indicative actually.

2-SAXS analysis.

The authors use SAXS to characterize the solution–extended conformation of TgGAC and show that it is different from the crystal conformation.

Reading Kumar et al. (2021) one notices that crystals were grown at pH=5 using a protein initially purified at pH=8.0.

There is a piece of data that could be shown by the Authors in that study. It would be the superimposed SEC profiles of purified TgGAC at pH 8 and pH 5 under a diluted regime maybe (but with some salt around) see my comment below.

Authors show SAXS profiles at different pHs in a diluted regime (Figure 2b), how is pH change performed? They notice that oligomerization seems the most pronounced at pH5 (the crystallization pH) but we will also make the observation that the SAXS buffer has virtually no salt (25 mM Tris) and these are not ideal conditions to limit intermolecular interactions and maybe aggregation independently of molecular compaction (closing of the structure as in the crystal conformation).

The P(r) compared are different. The solution curve was obtained at pH 8 but the calculated curve was done on a structure obtained at pH 5. The N/C interface being so charged it would not be surprising that it is sensitive to pH changes. In this case, why not measure and calculate the experimental P(r) function at pH5 would it fit (or not, like what is observed in pH 8) the crystal structure? The SAXS study of Jacot et al. (2016) mentions a range of concentrations from 1.2 to 4.6 mg/ml for the equivalent construct (FL–TgGAC) on a wider angular range it seems compared to what is described here.

Since the authors present P(r) analysis, in addition to the large difference in RG it would be meaningful to emphasize in the figures the maximum intramolecular distance (Dmax) calculated with the X–ray structure (~145 Å from what I see in Figure S3b/S2c) to the 400Å distance estimated by Fourier inversion of the experimental scattering curve. This clearly shows that the structure seems more elongated in solution providing that there is no significant aggregation or oligomerization.

Concerning the ab initio reconstruction approach, since the authors have a crystal structure could they not model a few subsets of extended structures using simple rigid body motions or the models later discussed in Fig8 and assess the quality of the corresponding I(q) fit with programs such as Crysol.

Figure S3a.

Small typo in the figure, it should be eight instead of eight and the X–axis unit legend is partially occulted for the scattering angle q2 (Å–2).

For the Guinier analysis. Fitting a straight line on a scattering curve is usually done indicating the corresponding q.RG range of points (only 8 points here based on what is reported).

Thus the q.RG range would be going up to ~ 0.017x105=1.8 which is rather generous but not unheard of. I understand this was done with a high–flux in–house instrument (described in Lyngso and Pedersen J. Appl. Cryst. 2021, 54), not on a synchrotron beamline. Could the distance between the sample and detector be specified and also the resulting angular range(s) used to measure in the Guinier (low q) and P(r) regimes (high q), were two curves merged or was it a single I(q) curve used for the Fourier inversion?

Figure S3b which is also Figure 2c? For the pair–distance distribution function P(r) derived from experimental scattering curves, it is unclear to me if the error bars are included in the graphs. The concentration range seems quite low to obtain a scattering curve with a good s/n ratio in the large q, high–resolution regime.

There is also a typo. MW should be 286 kDa instead of mg/ml

I agree overall with the conclusions of the SAXS section despite an issue with the role of pH and the buffer conditions used here. Besides the few errors that need to be corrected in the figures and some experimental details that could be added in the methods section.

Page 6. "collectively, it can be concluded that a closed structure…"

Would there be a way to illustrate this then? Using a sequence co–conservation projection coloring projected on the two surfaces (N/C) involved.

The PH domain of TgGAC binds to membranes enriched in PA through two basic patches as shown by nice MD simulations (Figures3 and S3) and NMR studies using also the corresponding PfGAC PH domain using elegant (at least to me) relaxation methods to map and somehow quantify interactions between PH domain and either nanodiscs or LUVs which they should define as large unilamellar vesicles somewhere in their manuscript although they also assimilate them to liposomes (Figures4, S4 and S5).

I would suggest the authors state the sequence conservation (similarity or identity) % based on PH structure–guided sequence alignments for Pf and Tg. Sequence conservation is mentioned in the introduction but not 'quantified' explicitly, as a matter of fact, the information would be useful for the full–length structure. This brings back my comments about my surface conservation analysis for Apicomplexa.

While Pf is obviously also of 'interest' (maybe more so than Tg) and it emphasizes conservation of the process throughout the phylum Apicomplexa, is there a practical/technical reason to jump between Tg and Pf systems for the PH–lipid interaction studies by NMR? It sometimes gets a bit confusing to jump back and forth.

The mapping of the PA binding site on the PH domain is well described and is validated by site–directed mutagenesis on isolated TgGAC–PH domain but also on full–length TgGAC (Figure 5). Although the liposome binding assay is a simple centrifugation assay not a classical flotation assay with gradient centrifugation separation in sucrose, the results are rigorously presented and support the conclusion about the role played by the PH domain during GAC preferential binding to PA–enriched membranes

Page 9. The title 'Targeted mutations in GAC PH domain are not fitness conferring in vivo' confuses me. Is conferring truly what the authors intended? I would think they meant targeted mutations do NOT affect fitness or virulence. Am I missing something?

The next two sections of the manuscript (but not the discussion) lack clarity and require some substantial rewriting/detailing to help the reader follow the intellectual thread developed by the Authors.

H/D exchange analysis and Figure 7 as a whole

While panels b and c are clear and show some endogenous interaction between TgGAC and endogenous parasitic actin. The rest of the section associated with Figure 7 needs some serious rewriting and explaining. What are GAC N70 and GAC N100? Despite figure S3 I genuinely do not understand.

I have absolutely no clue as to what I am looking at in Figures 7d and 7e and furthermore, panel 7e is not captioned either so please clarify this (at least to me) in the figure and the main text of the manuscript.

Also in Fig7 panel a. Something seems wrong or misleading between the helical cartoon model on the left and the schematic on the right. Why is there magenta on the top of the schematic? Is that the linker, if so it should be colored differently and in a thinner line, then the linear protein schematic below should be edited accordingly.

Page 10/11. Modelling of TgGAC binding to MIC2.

I would suggest precision (again) about what MIC2 is, some kind of adhesin. It is only briefly mentioned once in the introduction on page 3, but that is about it. Please help us (re)connect some dots there. Not everyone is an addict to glideosome machineries.

Figure 8 should be annotated more rigorously. In the legend.

8a left side should be captioned a bit more carefully and rigorously GAC ARII–ARIII it seems, the rest is omitted. MIC2 tail (red I guess).

The description becomes increasingly unclear and we are left trying to guess what we are looking at. Page 11.

De facto these are not superimpositions. Correct me if wrong.

You have a model (green GAC arch 39–42 + red MIC2 ?) of your complex (left panel) and shown next to them in the same relative orientation as the corresponding other structurally/functionally related complexes p120, catenin, etc to emphasize peptide/ARM repeats interactions. E–cadherin labels could be colored in red and it suddenly becomes crystal clear.

Panel 8b is similarly and annoyingly cryptic.

8b left side…what is the magenta–pink thing? An F actin filament…

8b right side…the boxed structure is a crystal or cryoEM structure of? At least a PDB code and a reference in the legend would be required.

The discussion is clearly written and puts things in perspective summarizing the roles of PH in PA binding and protein stabilization although this latter issue is mentioned but not really explained.

Since it seems that the PH domain is not required for fitness and membrane binding seems dispensable, the authors state that its presence is required for proper folding of the full–length construct… Has there been any attempt to generate parasites lacking the c–terminal PH domain?

I have one main objection. The oligomerization state induced by pH as seemingly observed by SAXS.

Having done a fair amount of X–ray and neutron scattering, I am somehow puzzled by the experimental buffer conditions chosen by the authors (the same applies to the study by Jacot et al. that I looked at). There is no salt present, even under the relatively low protein concentration described here it is reasonable to think that they could be non–specific interactions between molecules.

I would temper the interpretation of the data at pH 5 with that so–called oligomerization that could be aggregation in disguise.

I quote Kumar et al. in Crystals 2021

Reproducible protein crystals were obtained in 100 mM magnesium acetate, 100 mM sodium acetate, 6% PEG8000, and pH 5.0. These were manually optimised by screening over sodium acetate pH ranges of 4.0 to 5.0 in one dimension and a PEG8000 concentration gradient of 4%–10% in the second dimension. Crystallization was set up at a concentration of 5–60 mg mL−1.

Obviously, the protein must not be that unhappy at pH 5 to crystallize but it is in presence of some significant amount of soft salts.

I would really be curious to see a SEC of FL–GAC profile at ph 5 and 8 in presence of a decent amount of salt like we crystallographers like to use when purifying proteins. Ideally, the corresponding SAXS in more 'salty conditions' might be of interest.

The authors put their work in the perspective of understanding glideosome assembly and mechanism of action to promote adhesion and the generation of movement/gliding of parasites as they enter or egress the host cell. The availability of the first full-length atomic structure of an apicomplexan GAC is an important step towards a mechanistic understanding of glideosome assembly and action. Identification of protein-protein and protein–lipid interactions can provide new ways to design anti-parasitic drugs.

*Reviewer #3 (Recommendations for the authors):*

There are several areas in which the manuscript could be improved.

– A schematic diagram placing the GAC in the context of the parasite would be a valuable addition to figure 1, as would including the sequence diagram from Figure 7a in Figure 1.

– More information is required to judge the reproducibility of binding data in the relevant figure panels. How many times were binding experiments repeated? Do repeats represent technical repeats with the same protein preparation or different preparations? Do error bars represent SEM? Please add this information to relevant figure legends.

– Figure S1. It is difficult to judge concordance between experimental NOE data and NOE count from the AlphaFold model. Can a chart of experimental NOEs be shown for comparison? Panel E requires colour key.

– Sequence conservation of the N/C interface is mentioned in the text several times, but it would be useful to show this plotted on the structure.

– The AlphaFold prediction of the interface between GAC and MIC2 is interesting but not convincing in its current form. A PAE plot is required to judge the confidence of the GAC–MIC2 interface. pLDDT scores (ideally plotted on the structure) are required to judge confidence in the conformation of the MIC2 peptide.

– Does AlphaFold predict TgGAC to be fully closed, as in crystal structure, or partially separated, as in the PfGAC prediction?

– p9. The section titled "Targeted mutations in GAC PH domain are not fitness conferring in vivo" is confusing. Key lipid–binding residues in the GAC PH domain are not essential for parasite invasion.

– Figure 8b, middle panel. The basis for how the GAC is placed with respect to actin is unclear – it would seem that there are many other ways it could be placed that are equally compatible with the data.

– There are numerous typographical issues, including mis-numbering of the supplemental figures, which make the manuscript difficult to read.

---

## [Author Response]

Essential revisions:1) Although the structure of GAC is overall well described it requires further polishing in terms of model quality and accuracy (as it stands it appears like a structure at an early stage of refinement). Please carefully address the reviewer comments as detailed below in the recommendations for the authors.

The authors have been progressing the refinement and the current model is now significantly improved. See comments below and validation report.

2) Alternative interpretation of the SAXS data should be considered concerning the oligomerization state of GAC induced by pH.

We have toned down the discussion of the pH dependent SAXS data (section GAC adopts multiple extended conformations in solution). The choice of ‘no salt’ in our ideal buffer came out of numerous screens for GAC stability, homogeneous behaviour in size exclusion chromatography, reproducibility in crystallisation. We have we now included size exclusion profiles in Figure 2—figure supplement 1, which confirm oligomerisation at pH 5 and that salt has no effect of the behaviour of TgGAC at either pH.

3) The manuscript should be carefully reviewed to correct numerous errors in the text, figures and legends.

The manuscript has been thoroughly checked for textual mistakes and these have fixed.

4) The model presented in Figure 8 is too speculative and the figure should be removed.

The speculative models in Figure 8 have been removed. The structural comparison of the C-terminal arch of TgGAC with the β-catenin ARM structure is shown as a supplementary figure (Figure 1—figure supplement 3), without the modelled structure of TgGAC/MIC2 as that requires further validation. We now comment of the structural similarity of the β-catenins in the discussion (lines 424-429).

Reviewer #1 (Recommendations for the authors):– The crystal structure model a long way from being good enough in quality and appears like a structure at an early stage of refinement. It has a large number of Ramachandran and bond length and angle outliers. Looking at the model, there are also large areas in which the model does not fit the map. I have not mentioned this in the public report, to be kind, but this really must be fixed, or it will be embarrassing. A structure at this resolution should really have no outliers (except perhaps for a few Ramachandran outliers if the density shows them to be genuine) of these types and should fit the maps.

Agreed and we thank the reviewer for his expert analysis of the GAC crystal structure. The authors have been progressing the refinement and made significant modifications and improvements to the model. Individual residues that the reviewer kindly earmarked have been addressed. The Ramachandran outlier statistics have been improved (<1%). For some solvent exposed loop regions, the authors have attempted to model the residues as best possible, but a few Ramachandran outliers persist in those regions where the electron density is poor.

– Figure 1 does not make clear what is a model and what an experimentally derived structure is. I would show the NMR and crystal structures as the first two panels and only then the composite. In the era of AlphaFold, it is important to distinguish clearly between experimentally determined structures and models.

Suggested additional figure panels are included in Figure 1 to clearly separate the crystal structure and the NMR-structure/AlphaFold model of the PH domain. The Figure 1 caption now explained the full-length structure is a hybrid of both.

– There are quite a lot of errors in the text and figures and legends.

The manuscript has been thoroughly checked for textual mistakes and these have fixed.

– Error in Figure 4 legend – two (e) and no (f).

Fixed.

– Error in Figure 5 legend – there is no (d) or (e) in the figure.

Fixed.

– Comment on the differential effect of mutations – for example, KER – on full–length and PH domains.

Agreed this is interesting and we now comment on this and the implication for other regions in full-length GAC contributing to membrane binding (line 281-284).

– Figure 7d/e – what are GAC N100 and GAC N70? What is peptide centroid? Why is this not shown for the full protein length when GAC fl has been studied in this way? What is e? It felt to me that the data in Figure 7d/e would fit better with the SAXS data and make more sense to readers here when describing the flexibility of GAC.

Figure 7 and the H/D exchange section has been completely reworked to improve clarity and accuracy. In summary, the HDX-MS in Figure 7 establishes that the conformation of the TgGAC_1-619_ and TgGAC_1-1114_ have similar conformations to when in context of full-length TgGAC by probing dynamics and accessibility of amides to exchange. We also show a supplementary Figure S6c of the NMR spectrum for TgGAC_1-619_ indicating that it is well folded.

– Personally I would remove Figure 8 as it is extremely speculative.

The speculative models in Figure 8 have been removed. The structural similarity of the C-terminal arch of TgGAC with the β-catenin ARM structure is shown as a supplementary figure (Figure 1—figure supplement 3),, without the modelled structure of TgGAC/MIC2 as this requires further validation. We now comment of the structural similarity of the β-catenins in the discussion (line 426-430).

Reviewer #2 (Recommendations for the authors):1. Structure (X–ray and NMR) description.The full–length and quite large (2639 residues) structure was solved in a composite way. The largest part (7–2504) solved Se–MAD crystallography revealing 53 modules resembling armadillo/heat–like repeats and a small (2505–2639) and dynamic pleckstrin homology–like domain C–terminal domain solved by NMR assisted with AlphaFold2 modelling.The structure is well–described and illustrations are relatively clear to get a sense of the modular organization of this large modular protein. Figure S2 in particular is to be mentioned as the Authors took the time to generate a SS mapping of the entire large structure on the sequence.

Figure S2 (now Figure 1—figure supplement 2 is now explicitly mentioned in the text (line 115))

This reviewer is a crystallographer and inspected the density map in its entirety in COOT using the PDB and two maps provided 2Fo–Fc and Fo–Fc (thank you for the PDB and maps). It is a large protein so I am aware it represents a considerable amount of work but the repetitive modular organization is actually helpful to guide building (especially with other related structures at hand). This structure still requires some significant polishing in some areas where things can be improved and clearly cleaned up especially for this resolution range considering it was phased by MAD.

We thank the reviewer for this comment. Significant improvements have now been made to the model as stated in the response above.

The statistics for crystal structure resolution (i.e., Se-MAD phasing) are not reported accurately. This should be remedied.

We thank the reviewer for his observations on the statistics presented in the manuscript. A revised table has been provided to address the lack of phasing statistics. The improved refinement statistics are also provided.

Phasing and refinement.Data quality and refinement/phasing.Outer shell statistics with Rmerge and Rmeas of over 100% with an I/σ(I) of only 0.71 are a bit extreme even to me. I know that the contributions of weak reflections are weighted down with all the maximum likelihood refinement procedures but still. Please report the multiplicity/redundancy of the overall data set and the outer shell in this present version of the study.Table 1. Since no phasing information is reported there, it would be adequate to remove phasing from its title. It is only data quality and refinement statistics.That said it is expected to scientifically and rigorously report the phasing statistics since no phasing statistics can be found anywhere in this manuscript. I am sure the first and last authors will agree with me on this.

A revised table has been provided to address the lack of phasing statistics.

Refinement. In light of what I saw while inspecting the whole structure and ED Fourier difference maps, it might be useful to consider anisotropic TLS refinement of atomic displacement parameters, especially in light of the 'subdomain' structures suggested by SAXS/HDX and MD simulation data. Coloring the structure by the B factor on the backbone is quite indicative actually.

We thank the reviewer for their advice in refinement procedures. We have applied both mixed anisotropic TLS refinement parameters to our refinements and the R factors have improved (19% and 26%) as well as the resulting electron density maps.

2-SAXS analysis.The authors use SAXS to characterize the solution–extended conformation of TgGAC and show that it is different from the crystal conformation.Reading Kumar et al. (2021) one notices that crystals were grown at pH=5 using a protein initially purified at pH=8.0.There is a piece of data that could be shown by the Authors in that study. It would be the superimposed SEC profiles of purified TgGAC at pH 8 and pH 5 under a diluted regime maybe (but with some salt around) see my comment below.

We have included a supplementary figure of the SEC profiles (Figure 2—figure supplement 1A) which show that as expected TgGAC behaves significantly larger at pH 5 than pH 8, and salt does not influence the elution volume (lines 158-160).

Authors show SAXS profiles at different pHs in a diluted regime (Figure 2b), how is pH change performed? They notice that oligomerization seems the most pronounced at pH5 (the crystallization pH) but we will also make the observation that the SAXS buffer has virtually no salt (25 mM Tris) and these are not ideal conditions to limit intermolecular interactions and maybe aggregation independently of molecular compaction (closing of the structure as in the crystal conformation).

The pH was adjusted from pH 8.0 to the desired pH by addition of the appropriate amount of 1 M HCl, which gives rise to a maximum salt concentration of 30 mM at pH 4.0. This is now added to the Materials and methods section. As the increase in ionic strength increases the screening of intermolecular interactions, lowering the pH should decrease rather than increase the tendency towards oligomerization for GAC. We presume the oligomerization observed at lower pH must stem from conformation changes rather than this small difference in ionic strength. We present our SEC profiles at high/low pH and salt concentration and demonstrate and no beneficial effect of salt on elution.

The P(r) compared are different. The solution curve was obtained at pH 8 but the calculated curve was done on a structure obtained at pH 5. The N/C interface being so charged it would not be surprising that it is sensitive to pH changes. In this case, why not measure and calculate the experimental P(r) function at pH5 would it fit (or not, like what is observed in pH 8) the crystal structure?

It is true that the p(r) functions are from, respectively, a structure at pH 5.0 and solution data at pH 8.0. Ideally, we would like to compare the structures at pH 5.0 in solution with the crystal structure, but as GAC oligomerises/aggregates at pH 5.0, we are not able to determine an overall size accurately. The absence of aggregation and the ability to determine overall size is a prerequisite for calculating the p(r) as one needs to estimate the D_max_. We have now stressed this point in the manuscript to make it clearer. We compare SAXS data at two pH values to demonstrate the presence of a much larger (aggregated) structure at a higher pH relative to the crystal structure.

The SAXS study of Jacot et al. (2016) mentions a range of concentrations from 1.2 to 4.6 mg/ml for the equivalent construct (FL–TgGAC) on a wider angular range it seems compared to what is described here.

We measure at relatively low concentration (1.3 mg/mL), as we want to go as low as possible to avoid any affect from intermolecular interactions that could promote aggregation. On the other hand, we cannot go significantly lower as the statistics on the data will then be considerable worse, so 1.3 mg/mL turned out to be the best compromise. We have added a comment about this in the Results section “GAC adopts multiple extended conformations in solution”. SAXS data in Jacot et al. (2016) are shown as a function of *s* (equivalent to *q*) on a linear scale, where we show data on a logarithmic s/q scale. We believe a logarithmic scale is more suitable as it more clearly shows the data at low *q*, where one would see effects from intermolecular interactions. On a linear s/q scale, this region is very small, and these important effects can be more difficult to detect. As data are on either a logarithmic or linear scale, it is not straightforward to compare the scales. Jacot et al. (2016) do not give a *q* range in the article, but we have copied the figure and estimated the position of the first and last point, which give a *q* range of around 0.012 – 0.43 Å^-1^ (see Author response image 1 for estimation of first and last point). Apparently, the two curves at the top have a larger range, however, this is in fact extrapolated model fit curves and not data. In our data we have a *q* range of 0.0098 – 0.42 Å^-1^ (which is also mentioned in the Materials and methods section now), so the range is similar to Jacot et al. (2016).

**Author response image 1. sa2fig1:** 

Since the authors present P(r) analysis, in addition to the large difference in RG it would be meaningful to emphasize in the figures the maximum intramolecular distance (Dmax) calculated with the X–ray structure (~145 Å from what I see in Figure S3b/S2c) to the 400Å distance estimated by Fourier inversion of the experimental scattering curve. This clearly shows that the structure seems more elongated in solution providing that there is no significant aggregation or oligomerization.

Thank you for pointing this out. We have now emphasized this in the text to make it clearer (section ‘GAC adopts multiple extended conformations in solution’).

Concerning the ab initio reconstruction approach, since the authors have a crystal structure could they not model a few subsets of extended structures using simple rigid body motions or the models later discussed in Fig8 and assess the quality of the corresponding I(q) fit with programs such as Crysol.

We have tried modelling extended structures using simple rigid body motions and we devised another approach in which steered-MD was used with the application of a pulling force between residues on either side of the interface to drive its separation and the extension of the structure. The structure that fit best was the most extended generated from MD. These fits are presented in Figures 3 and described in the SAXS section ‘GAC adopts multiple extended conformations in solution’.

For the Guinier analysis. Fitting a straight line on a scattering curve is usually done indicating the corresponding q.RG range of points (only 8 points here based on what is reported).Thus the q.RG range would be going up to ~ 0.017x105=1.8 which is rather generous but not unheard of.

We have now indicated it on the structure. It is 0.0155 Å^-1^ x 105 Å = 1.6.

I understand this was done with a high–flux in–house instrument (described in Lyngso and Pedersen J. Appl. Cryst. 2021, 54), not on a synchrotron beamline. Could the distance between the sample and detector be specified and also the resulting angular range(s) used to measure in the Guinier (low q) and P(r) regimes (high q), were two curves merged or was it a single I(q) curve used for the Fourier inversion?

Yes, the distance is 867 mm, the *q* range is 0.0098 – 0.42 Å^-1^, and one single curve was used where the whole *q* range could be probed in one measurement. This information is available in the article references for more information about the instrument, but we have now also added it to the Materials and methods section.

Figure S3b which is also Figure 2c? For the pair–distance distribution function P(r) derived from experimental scattering curves, it is unclear to me if the error bars are included in the graphs. The concentration range seems quite low to obtain a scattering curve with a good s/n ratio in the large q, high–resolution regime.

Error bars were not included initially but they are now and are small. We chose as low a protein concentration as possible while maintaining a good signal to noise ratio. Usually, such good statistics cannot be obtained with a concentration of 1.3 mg/mL, but keep in mind that GAC is a very large protein (286 kDa) and that the intensity scales with protein mass, so therefore much better data can be obtained at low concentrations for GAC compared to proteins with masses of 30-50 kDa.

I agree overall with the conclusions of the SAXS section despite an issue with the role of pH and the buffer conditions used here. Besides the few errors that need to be corrected in the figures and some experimental details that could be added in the methods section.

Fixed.

Page 6. "collectively, it can be concluded that a closed structure…"Would there be a way to illustrate this then? Using a sequence co–conservation projection coloring projected on the two surfaces (N/C) involved.

Fixed and we now include a surface representation of the interface with Consurf conservation scoring in Figure 2b and a full sequence alignment in Figure 1—figure supplement 2.

The PH domain of TgGAC binds to membranes enriched in PA through two basic patches as shown by nice MD simulations (Figures3 and S3) and NMR studies using also the corresponding PfGAC PH domain using elegant (at least to me) relaxation methods to map and somehow quantify interactions between PH domain and either nanodiscs or LUVs which they should define as large unilamellar vesicles somewhere in their manuscript although they also assimilate them to liposomes (Figures4, S4 and S5).

LUVs and their size are now defined. Further details of LUV and nanodisc preparations are described (line 221 and in the Material and Methods).

I would suggest the authors state the sequence conservation (similarity or identity) % based on PH structure–guided sequence alignments for Pf and Tg. Sequence conservation is mentioned in the introduction but not 'quantified' explicitly, as a matter of fact, the information would be useful for the full–length structure. This brings back my comments about my surface conservation analysis for Apicomplexa.

This is now stated (53% identity) in the appropriate section on line 217. We also include a full annotated sequence alignment as Figure 1—figure supplement 2.

While Pf is obviously also of 'interest' (maybe more so than Tg) and it emphasizes conservation of the process throughout the phylum Apicomplexa, is there a practical/technical reason to jump between Tg and Pf systems for the PH–lipid interaction studies by NMR? It sometimes gets a bit confusing to jump back and forth.

Yes, it was a practical and technical reason for the Tg/Pf switch for the NMR-based lipid binding studies. To summarise, our NMR assignment for TgGAC_PH,_ was incomplete with several surface exposed amide NMR resonances between residues 2548-2560 absent from spectra. While this did not prevent our NMR structure validation, we felt that potential interacting residues could be missed in titrations with TgGAC_PH_. We therefore first explored PfGAC_PH_ which displayed improved NMR spectra and near complete assignment was possible, which facilitated a comprehensive map of the PA binding surface. These results were then confirmed and validated with equivalent NMR experiments on TgGAC_PH._ We have rewritten this explanation to improve clarity (from line 214).

Page 9. The title 'Targeted mutations in GAC PH domain are not fitness conferring in vivo' confuses me. Is conferring truly what the authors intended? I would think they meant targeted mutations do NOT affect fitness or virulence. Am I missing something?

This has been rephrased to remove confusion.

The next two sections of the manuscript (but not the discussion) lack clarity and require some substantial rewriting/detailing to help the reader follow the intellectual thread developed by the Authors.

These sections has been rewritten to improve clarity and comprehension (sections Evaluation of GAC and GAC fragments binding to toxoplasma F-actin and Investigation of GAC fragment conformations using hydrogen/deuterium exchange coupled to mass spectrometry).

H/D exchange analysis and Figure 7 as a whole.

Figure 7 and the H/D exchange section has been reworked to improve clarity and accuracy. In summary, the HDX-MS in Figure 7 establishes that the conformation of the TgGAC_1-619_ and TgGAC_1-1114_ have similar conformations to when in context of full-length TgGAC by probing dynamics and accessibility of amides to exchange. We also show a supplementary Figure 7—figure supplement 1C of the NMR spectrum for TgGAC_1-619_ indicating that it is well folded.

Page 10/11. Modelling of TgGAC binding to MIC2.I would suggest precision (again) about what MIC2 is, some kind of adhesin. It is only briefly mentioned once in the introduction on page 3, but that is about it. Please help us (re)connect some dots there. Not everyone is an addict to glideosome machineries.

TgMIC2 is now introduced properly within the introduction (from line 45) and refreshed where mentioned in the discussion.

Figure 8 should be annotated more rigorously. In the legend.8a left side should be captioned a bit more carefully and rigorously GAC ARII–ARIII it seems, the rest is omitted. MIC2 tail (red I guess).The description becomes increasingly unclear and we are left trying to guess what we are looking at. Page 11.

There is significant consensus on the speculative nature of Figure 8, so it has been removed as suggested. We have however included the structural comparison of the C-terminal arch of TgGAC with the β-catenin ARM structure in a supplementary figure S7, without the modelled structure of TgGAC/MIC2 as that requires further validation. We now comment of the structural similarity with the β-catenins in the discussion (paragraph starting line 415).

De facto these are not superimpositions. Correct me if wrong.

See above point, the legend in the now Figure 1—figure supplement 3has been reworded accordingly.

You have a model (green GAC arch 39–42 + red MIC2 ?) of your complex (left panel) and shown next to them in the same relative orientation as the corresponding other structurally/functionally related complexes p120, catenin, etc to emphasize peptide/ARM repeats interactions. E–cadherin labels could be colored in red and it suddenly becomes crystal clear.

See above point.

Panel 8b is similarly and annoyingly cryptic.8b left side…what is the magenta–pink thing? An F actin filament…8b right side…the boxed structure is a crystal or cryoEM structure of? At least a PDB code and a reference in the legend would be required.

See above point.

The discussion is clearly written and puts things in perspective summarizing the roles of PH in PA binding and protein stabilization although this latter issue is mentioned but not really explained.Since it seems that the PH domain is not required for fitness and membrane binding seems dispensable, the authors state that its presence is required for proper folding of the full–length construct… Has there been any attempt to generate parasites lacking the c–terminal PH domain?

Yes, there has been multiple attempts to generate GAC-ΔPH in the parasites either expressed as second copy or by deletion in the endogenous locus but all have failed. Expression appears to be very low, and the mutation is lethal. Concordantly, recombinant expression of GAC-ΔPH in bacterial/insect cells also failed suggesting that the mutant protein is not folded properly and unstable. This prompted us to focus on point mutants instead. See Author response image 2 for reviewer information.

**Author response image 2. sa2fig2:** Production and purification of recombinant GAC-FL and GAC-∆PH. Both GAC contract constructs are well expressed in BL-21 bacteria, showing an intense band at the expected MW. Purification using a combination of affinity, anion-exchange and size-exclusion chromatography (SEC) shows excellent result for GAC-FL. Although some protein elultes in the void volume 8ml on Superdex200 SEC, most of GAC-FL comes off the column as a symmetrical peak at 9.7ml suggested of monomeric protein. Purification of GAC-∆PH following a similar procedure as for GAC-FL shows many degradation products and absence of the intact protein on the SEC column. The purifications described here are representative of several attempts with no GAC-∆PH protein that could be isolated. The results indicate that C-terminal PH domain is essential for recombinant GAC stability, with removal of the C-terminal PH domain leading to degradation of GAC.

I have one main objection. The oligomerization state induced by pH as seemingly observed by SAXS.Having done a fair amount of X–ray and neutron scattering, I am somehow puzzled by the experimental buffer conditions chosen by the authors (the same applies to the study by Jacot et al. that I looked at). There is no salt present, even under the relatively low protein concentration described here it is reasonable to think that they could be non–specific interactions between molecules.I would temper the interpretation of the data at pH 5 with that so–called oligomerization that could be aggregation in disguise.

The buffer choice came out of numerous screens for GAC stability, homogeneous behaviour in size exclusion chromatography, reproducibility in crystallisation and we concluded that high pH, low or no salt and reducing agent is where GAC behaved best. Agreed the low pH behaviour this could be an in vitro artifact of aggregation, so the discussion has been toned down.

Reviewer #3 (Recommendations for the authors):There are several areas in which the manuscript could be improved.– A schematic diagram placing the GAC in the context of the parasite would be a valuable addition to figure 1, as would including the sequence diagram from Figure 7a in Figure 1.

This is now included.

– More information is required to judge the reproducibility of binding data in the relevant figure panels. How many times were binding experiments repeated? Do repeats represent technical repeats with the same protein preparation or different preparations? Do error bars represent SEM? Please add this information to relevant figure legends.

These are now included in caption and methods.

– Figure S1. It is difficult to judge concordance between experimental NOE data and NOE count from the AlphaFold model. Can a chart of experimental NOEs be shown for comparison? Panel E requires colour key.

We took the opportunity to calculate an NMR structure from our available NOESY data and made a comparison with the Alphafold model. Figure S1 has been modified to show the calculated NMR ensemble and illustrate the comparison (paragraph from line 91)

– Sequence conservation of the N/C interface is mentioned in the text several times, but it would be useful to show this plotted on the structure.

We now include a surface representation of the interface with Consurf conservation scoring in Figure 2b and a full sequence alignment in Figure 1—figure supplement 2.

– The AlphaFold prediction of the interface between GAC and MIC2 is interesting but not convincing in its current form. A PAE plot is required to judge the confidence of the GAC–MIC2 interface. pLDDT scores (ideally plotted on the structure) are required to judge confidence in the conformation of the MIC2 peptide.

The confidence scores are reasonable for the MIC2 peptide however due to it speculative nature we now removed Figure 8 as suggested, but keep the basic comparison of the C-terminal ARM arch of GAC with the catenins and display this in Figure 1—figure supplement 3.

– Does AlphaFold predict TgGAC to be fully closed, as in crystal structure, or partially separated, as in the PfGAC prediction?

We have calculated an Alphafold model for TgGAC and like PfGAC the N/C interface is also partially separated. This is now shown together in Figure 2.

– p9. The section titled "Targeted mutations in GAC PH domain are not fitness conferring in vivo" is confusing. Key lipid–binding residues in the GAC PH domain are not essential for parasite invasion.

The title has been reworded as suggested.

– Figure 8b, middle panel. The basis for how the GAC is placed with respect to actin is unclear – it would seem that there are many other ways it could be placed that are equally compatible with the data.

See above but the figure has removed.